# Phosphorylated histone variant γH2Av is associated with chromatin insulators in *Drosophila*

**James R. Simmons**[ID]<sup></sup>, **Ran An**<sup>†</sup>, **Bright Amankwaa**, **Shannon Zayac**, **Justin Kemp**, **Mariano Labrador**[ID]*

Department of Biochemistry and Cellular and Molecular Biology, The University of Tennessee, Knoxville, Tennessee, United States of America

☯ These authors contributed equally to this work.
† Deceased.
* labrador@utk.edu

**Data Availability Statement:** All original microscopy image data files and numerical data that underlies graphs and statistical analysis in this work are available at the DRYAD data repository

## Abstract

Chromatin insulators are responsible for orchestrating long-range interactions between enhancers and promoters throughout the genome and align with the boundaries of Topologically Associating Domains (TADs). Here, we demonstrate an association between *gypsy* insulator proteins and the phosphorylated histone variant H2Av (γH2Av), normally a marker of DNA double strand breaks. *Gypsy* insulator components colocalize with γH2Av throughout the genome, in polytene chromosomes and in diploid cells in which Chromatin IP data shows it is enriched at TAD boundaries. Mutation of insulator components *su(Hw)* and Cp190 results in a significant reduction in γH2Av levels in chromatin and phosphatase inhibition strengthens the association between insulator components and γH2Av and rescues γH2Av localization in insulator mutants. We also show that γH2Av, but not H2Av, is a component of insulator bodies, which are protein condensates that form during osmotic stress. Phosphatase activity is required for insulator body dissolution after stress recovery. Together, our results implicate the H2A variant with a novel mechanism of insulator function and boundary formation.

## Author summary

The DNA in eukaryotic genomes is folded into domains called Topologically Associating Domains (TADs), which promote gene specific transcription regulation. Insulator proteins are DNA binding proteins that bind at the boundaries between adjacent TADs. Loop extrusion is a mechanism by which insulators promote TAD formation in vertebrate genomes, but the mechanism by which insulator proteins facilitate the formation of boundaries in *Drosophila* is not well understood. In this work we show that there is an association between *Drosophila gypsy* insulator proteins and the phosphorylated version of the histone variant H2Av (γH2Av). γH2Av has been traditionally linked to the mechanism of DNA repair, but our data shows that *gypsy* insulator components colocalize with γH2Av throughout the genome, and that γH2Av is also enriched at TAD boundaries.

(https://doi.org/10.5061/dryad.k98sf7m8x) All
ChIP seq data analzed in this work was initially
published elsewhere. All ChIP-seq datasets are
available at the following NCBI GEO: https://www.
ncbi.nlm.nih.gov/geo/query/acc.cgi?acc=
GSM1900413; SRX1299941 (γH2Av; Li et al.,
2016) https://www.ncbi.nlm.nih.gov/geo/query/
acc.cgi?acc=GSM685610; SRX046654 (Su(Hw);
Chen et al., 2012), https://www.ncbi.nlm.nih.gov/
geo/query/acc.cgi?acc=GSM1001885; SRX186113
(Mod(mdg4)67.2; Matzat et al., 2012). https://
www.ncbi.nlm.nih.gov/geo/query/acc.cgi?acc=
GSM539583; SRX019957 (homotypic H2Av;
Weber et al., 2010) https://www.ncbi.nlm.nih.gov/
geo/query/acc.cgi?acc=GSM539579; SRX019953
(heterotypic H2Av; Weber et al., 2010), http://
chorogenome.ie-freiburg.mpg.de/data_sources.
html. TAD-separation scores (Ramirez et al., 2018).

**Funding:** This work was supported by a grant from
the National Institutes of Health (https://www.nimh.
nih.gov/) (MH108956). The funders had no role in
study design, data collection and analysis, decision
to publish, or preparation of the manuscript.

**Competing interests:** The authors have declared
that no competing interests exist.

Mutation of genes encoding insulator proteins *su(Hw)* and Cp190 results in a significant
reduction in γH2Av levels, and inhibition of the phosphatase activity that removes phos-
phate from γH2Av strengthens the association between insulator proteins and γH2Av.
We also show that γH2Av is a component of insulator bodies, which are protein conden-
sates that form during osmotic stress. Together, our results implicate the H2A variant
with a novel mechanism of insulator function and boundary formation.

## Introduction

Chromatin insulators were first characterized in *Drosophila* as a class of protein/DNA com-
plexes associated to specific sequences in the genome that work through two general functions:
to restrict communication between enhancers and promoters through physical separation into
different genomic domains and to prevent the spread of heterochromatin into euchromatic
regions of the genome [1–5]. The presence of insulators in the genome is conserved among
eukaryotes, with the CTCF insulator being the only known insulator protein in the human
genome [6]. Recently, insulator proteins have been shown to play a role in 3D-genome organi-
zation by facilitating the establishment of topologically associating domains (TADs) and are
often found enriched at TAD boundaries [7,8].

*Drosophila melanogaster* has an array of different insulator complexes, with each complex
being recruited to different sequences in the genome by a DNA binding insulator protein
[9,10]. One insulator site, located within the *gypsy* retrotransposon, has been thoroughly char-
acterized for its ability to block enhancer promoter communications [4]. A number of *gypsy*
retrotransposons are present throughout the *Drosophila* genome [11], and insertion or trans-
position of *gypsy* to a new locus may interrupt local transcriptional activity and chromatin
dynamics [4,12]. Insulator proteins are recruited to *gypsy* through a 460-bp sequence com-
posed of 12 binding sites for Suppressor of Hairy Wing (Su(Hw)) [4]. Su(Hw) specifically
recruits an isoform of *modifier of mdg4* (Mod(mdg4)67.2) [13]. Another protein, Centrosomal
Protein 190 (CP190), is found as an essential part of different insulator complexes [9] and is
recruited to the *gypsy* insulator through interactions with Mod(mdg4)67.2 [14] and the amino
terminal domain of HIPP1 (HP1 and insulator partner protein 1) [15].

Insulator-binding proteins in *Drosophila* can form aggregates known as insulator bodies
[16]. The role of these bodies in genome organization has been debated, and functions for
insulator bodies have been proposed from genome organization hubs to passive storage cen-
ters for insulator proteins [17–19]. Previous work in our lab has demonstrated a role for insu-
lator bodies in the cellular response to osmotic stress, with insulator proteins leaving
chromatin and forming bodies in the nucleoplasm as the environment becomes more hyper-
tonic [20]. Of the *gypsy* insulator proteins, Su(Hw) is perhaps the best-characterized. Mutation
of *su(Hw)* is associated with female infertility [21–24] and alters the cell's response to DNA
damage [25]. Outside of the *gypsy* insulator, Su(Hw) binds many sites alone or in conjunction
with Mod(mdg4)67.2 and CP190 [9,10,26–29].

Su(Hw) also participates in the DNA damage response, possibly as part of the search for
homologous sequences during homologous recombination [25]. A role for insulators in
homologous recombination-based DNA repair has been well established with mammalian
CTCF, which is recruited to sites of DNA double strand breaks (DSBs) [30–32]. One of the
first steps in the cellular response to DSBs is the phosphorylation of a variant of H2A known as
H2AX in mammalian systems and H2Av in *Drosophila* [33,34]. H2AX is phosphorylated by
ATM (Ataxia-telangiectasia-mutated) kinase and DNA-dependent protein kinase (DNA-PK)

in response to ionizing radiation [35] and by ATR (ataxia telangiectasia and Rad3-related) kinase after cells experience replication-induced genotoxic stress [36]. Phosphorylation of H2AX leads to recruitment of numerous proteins involved in the DNA damage response [37]. Upon resolution of the DSB, H2AX is dephosphorylated primarily by PP2A [38].

Our laboratory previously demonstrated an accumulation of phosphorylated H2Av (γH2Av) signal in the ovaries of *su(Hw)* mutants and the presence of chromosomal aberrations in actively dividing larval neuroblasts lacking Su(Hw), suggesting a possible connection between Su(Hw) activity and genome stability [24]. Furthermore, disruption of Mei-41/ATR, a kinase responsible for phosphorylating H2Av among other targets upon DNA damage [39], partially rescues the defective oogenesis phenotype associated with mutation of *su(Hw)* [24]. While insulator-binding proteins have been described for their role in genome organization and regulation in *Drosophila*, the mechanisms linking their activity to DNA repair remain elusive.

In this work, we show that γH2Av is present at Su(Hw)-binding sites throughout the genome, including at *gypsy* retrotransposons, and that mutation of several *gypsy* insulator components disrupts normal distribution of H2Av in chromosomes. We show that γH2Av is a component of insulator bodies formed under osmotic stress and that dephosphorylation of γH2Av is required for efficient dissolution of these bodies during recovery. Chromatin immunoprecipitation data reveal extensive genome-wide colocalization between Su(Hw) and γH2Av and enrichment for both at TAD boundaries. This association also extends to insulator function as flies doubly heterozygous for *His2Av^810* and mutant alleles of *su(Hw)* showed a partial rescue of phenotypes for *yellow^2* and *cut^6*, two *gypsy* insulator induced phenotypes. Collectively, these findings point to a model in which γH2Av works with insulators to coordinate genome function and perhaps genome-wide responses to genotoxic stress.

## Materials and methods

### Fly stocks and husbandry

All stocks were maintained on a standard cornmeal agar fly food medium supplemented with yeast at 20˚C; crosses were carried out at 25˚C. The following stocks are maintained in our lab and were originally obtained from Victor Corces (Emory University): *y^2w^1ct^6*; *cp190^H31-2*/ TM6B, *Tb^1*, *y^2w^1ct^6*; *cp190^P11*/TM6B, *Tb^1*, *y^2w^1ct^6*; *w^1118*; *su(Hw)^V*/TM6B, *Tb^1*, *y^2w^1ct^6*; *mod (mdg4)^u1*/TM6B, *Tb^1*. The stock *w^1118*; PBac(RB)*su(Hw)^e04061*/TM6B, *Tb^1* was obtained from the Bloomington *Drosophila* stock center (BDSC: #18224). The stock *y^2w^1ct^6*; *His2Av^810*/ TM6B; *Tb^1* was generated in our lab using *w^\**; *His2Av^810*/TM3; *Sb^1* from the BDSC (#9264). Our lab also provided the stocks Oregon R (OR) and *y^2w^1ct^6*; PBac(RB)*su(Hw)^e04061*/TM6B, *Tb^1* (derived from BDSC: #18224).

### Antibodies

Rabbit polyclonal IgG antibodies against Su(Hw), Mod(mdg4)67.2, and CP190 were previously generated by our lab [20,40]. A rat polyclonal IgG antibody against Su(Hw) generated by our lab was also used. Antibody against the phosphorylated form of H2Av (UNC93-5.2.1) [41] was obtained from the Developmental Studies Hybridoma Bank, created by the NICHD of the NIH and maintained at The University of Iowa, Department of Biology, Iowa City, IA 52242. These antibodies were all diluted 1:1 in glycerol (Fisher Scientific, BP229-1, lot 020133) and used at a final dilution of 1:200. Antibody against H2Av were purchased from Active Motif (Catalog #39715) and was used at a 1:200 final dilution. Secondary antibodies were all diluted 1:1 in glycerol and used at a final dilution of 1:200. The following secondary antibodies were used in this study: Alexa Fluor 594 goat anti-rabbit (Invitrogen, A-111037, lot 2079421), Alexa

Fluor 488 donkey anti-rabbit (Invitrogen, A-21206, lot 1834802), Alexa Fluor 488 goat anti-guinea pig (Invitrogen, A-11073, lot 84E1-1), Texas red donkey anti-rat (Jackson ImmunoResearch Laboratories, 712-075-150), and Alexa Fluor 488 goat anti-mouse (Invitrogen, A-11001, lot 1858182).

## Immunostaining of larval tissues

Wandering third instar larvae were dissected in PBS. Tissues were immediately placed into fixative (4% para-formaldehyde (Alfa Aesar, 43368, lot N13E011), 50% glacial acetic acid (Fisher Scientific, A38-212, lot 172788)) on a coverslip for one minute. Samples were squashed by lowering a slide on top of the sample then turning it over, placing it between sheets of blotting paper, and hitting the coverslip firmly with a small rubber mallet. Slides were dipped in liquid nitrogen, coverslips were removed, and samples were incubated in blocking solution (3% powdered nonfat milk in PBS + 0.1% IGEPAL CA-630 (Sigma-Aldrich, 18896, lot 1043)) for 10 minutes at room temperature. The slides were dried and incubated with primary antibodies overnight at 4˚C in a box humidified with wet paper towels. The next day, slides were washed twice in PBS + 0.1% IGEPAL CA-630 before incubation with secondary antibodies for three hours in the dark at room temperature. Slides were washed twice in PBS + 0.1% IGEPAL CA-630, treated with DAPI solution of 0.5 μg/mL (Thermo Fisher, D1306) for one minute, and washed one more time in PBS alone.

Samples were mounted with Vectashield antifade mounting medium (Vector Laboratories, H-1000, lot ZF0409) and coverslips were sealed with clear nail polish. All microscopy for immunostaining was performed on a wide-field epifluorescent microscope (DM6000 B; Leica Microsystems) equipped with a 100x/1.35 NA oil immersion objective and a charge-coupled device camera (ORCA-ER; Hamamatsu Photonics). Image acquisition was performed using Simple PCI (v6.6; Hamamatsu Photonics). Image manipulation was performed in FIJI [42]; all contrast adjustments are linear. Images were further processed in Adobe Photoshop CS5 Extended, Version 12.0 x64. Figures were assembled in Adobe Illustrator CS5, Version 15.0.0. Statistical analyses were performed in GraphPad Prism version 8.0.0 (224) (GraphPad Software, San Diego, CA).

## Immunostaining of S2 cells

For normal control conditions, S2 cells were incubated in insect medium (HyClone SFX-Insect Cell Culture Media; Fisher Scientific, SH3027802) supplemented with penicillin (50 units/mL) and streptomycin (50 μg/mL) (Gibco, 15070063) at 25˚C. To induce osmotic stress, the isotonic media was replaced with hypertonic media supplemented with 250 mM NaCl (Fisher Scientific, BP358-212). Cells were treated in this hypertonic stress media for 30 minutes. Coverslips were pretreated with pure ethanol (Decon Labs, 2716) and coated with concanavalin A (Sigma-Aldrich, C5275) to help S2 cells adhere to the glass surface. Cells were pipetted onto treated coverslips and were allowed to spread and adhere for 30 minutes. After treatment, cells were fixed (4% para-formaldehyde, 50% acetic acid) for 10 minutes at room temperature, followed by three washes with PBS buffer. Fixed cells were permeabilized with 0.2% Triton X-100 (Fisher Scientific, BP151, lot 014673) for five minutes then washed twice with PBS buffer. Cells were incubated in blocking solution (3% powdered nonfat milk in PBS + 0.1% IGEPAL CA-630) for 10 minutes at room temperature. Primary antibodies were diluted in blocking solution and samples were incubated in antibody solution overnight at 4˚C in a box humidified with wet paper towels. Unbound antibodies were washed off three times with PBST buffer (0.1% Triton-X 100). Secondary antibody incubation, DAPI staining, and mounting were performed as described above.

## Okadaic acid treatment

For the isotonic control samples, S2 cells were cultured in HyClone SFX-Insect media as above and incubated in 50 nM okadaic acid (Sigma-Aldrich, O9381) for 30 minutes. The hypertonic samples were obtained by shifting S2 cells from isotonic insect media to hypertonic conditions as described above and incubating for 25 minutes. After this, the hypertonic media was supplemented with 50 nM okadaic acid and cells were incubated for five more minutes. The isotonic recovery sample was obtained by first inducing hypertonic stress for thirty minutes, including okadaic acid for the final five minutes as above, then washing out the hypertonic media twice with isotonic media containing 50 nM okadaic acid. Cells were incubated in isotonic recovery media with okadaic acid for thirty minutes. Control samples were collected throughout the process following the same protocol without addition of okadaic acid. For the polytene chromosome example, salivary glands were dissected from wandering third instar larvae and incubated in 50 nM okadaic acid for 30 minutes before fixation and squashing.

## Fluorescence intensity and colocalization analysis

Images were analyzed for the amount of each protein (i.e. the intensity of each channel) using a macro script in FIJI [42]. The DAPI channel was used to automatically generate non-biased Region of interest (ROIs) for each cell, including all polytene chromosomes, which were then manually curated for extra precision. A rolling-ball background subtraction algorithm was used for all images. Intensity measurements were made using the measure function. Numerous images of polytene chromosomes were collected from each salivary gland squash. All acquisition parameters were kept constant between slides within each experiment. Intensities were compared using ANOVA analysis as indicated, with the choice of ANOVA tests based on the distribution of each data set.

Colocalization was quantified using the Coloc2 plugin in FIJI. This analysis uses the Costes method [43] to determine appropriate thresholds for each channel. Results are reported in terms of Pearson's Correlation Coefficient (PCC) [44], which describes the covariance between the two signals and ranges from -1 to +1, with positive numbers describing direct correlation between signal intensities, negative numbers representing anti-correlation of the signals, and zero representing no correlation between signals (i.e. random covariance) [45,46]. Another metric of colocalization is the Manders' Colocalization Coefficients (MCC) [45,47,48] for each channel–this relates how much of the signal in the green channel overlaps with signal in the red channel (M1) and how much of the signal in the red channel overlaps with signal in the green channel (M2). M1 and M2 may vary from 0, representing no overlap between signals, to 1, representing total overlap.

## ChIP-seq and TAD data analysis

The following NCBI publicly available ChIP-Seq datasets were used for the genome wide comparisons: SRX1299941 (yH2Av SRA [49]), SRX046654 (Su(Hw) SRA [50]), SRX186113 (Mod (mdg4)67.2 SRA [51]). The sequencing data was uploaded to the Galaxy web platform, and the public server at usegalaxy.org was used for analysis [52]. Briefly, the FastQ datasets from NCBI were mapped with Bowtie2 to produce BAM files [53]. Duplicate and unmapped reads were filtered out with SAM tools. Peaks were called with MACS2 callpeak, a Model-based analysis of ChIP-Seq [54], using Galaxy. We used SeqMINER version 1.3.4 for the downstream plotting analysis [55,56].

To compare the distribution of γH2Av with that of nucleosomal H2Av we used the high-resolution distribution of homotypic and heterotypic *Drosophila* H2Av nucleosomes from S2 cells obtained from a previous study [57]. FastQ datasets from paired-end reads from native

micrococcal nuclease–digested chromatin, enriched in homotypic H2Av (SRX019957 [57]) or heterotypic H2Av (SRX019953 [57]), were mapped with Bowtie2 to produce BAM files mapped to the dm6 *Drosophila* genome. We applied Bamcoverage to BAM files to generate H2Av Hom and H2Av Het bigwig files. SeqMINER was used to generate displays of the distribution of mean read density profiles (tags / 50 bp), using BED files to provide reference coordinates (± 10,000 bp). Bigwig files were generated by applying Bamcoverage to BAM files. The IGV genome browser (igv.org/app/) [58], using *Drosophila* dm6 or dm3 as the reference genome was utilized to visualize peak profiles and generate peak profile figures. The genomic distribution of TADs in the *Drosophila* genome [59] was used to produce a BED file generated with all genomic 1 kb fragments containing a TAD boundary at the center. When necessary, we used UCSC Genome Browser to convert dm3 coordinates into dm6.

To compare the distribution of γH2Av with the distribution of TAD-separation scores we applied DeepTools' multibigwigSummary (Galaxy Version 3.5.1.0.0) using γH2Av bigwig files obtained as described above and using 10,000 bp bin size. A bigwig file for TAD-separation scores was obtained from http://chorogenome.ie-freiburg.mpg.de/data_sources.html [59]. MultibigwigSummary analysis was performed using bigwig files mapped to the *Drosophila* dm3 reference genome. Spearman correlation coefficients were calculated using Plotcorrelation (Galaxy Version 3.5.1.0.0) on bin counts obtained from the multibigwigSummary analysis of the γH2Av and TAD-separation score distributions. Scatterplots for γH2Av and TAD-separation scores distribution were generated using Scatterplot (Galaxy Version 1.0.3).

## Western Blot

Wandering Oregon R (OR) third instar larvae were collected and homogenized in RIPA buffer (150 mM NaCl, 1% NP-40, 0.5% sodium deoxycholate, 0.1% SDS, 50 mM Tris-HCl, pH 8.0) supplemented with protease inhibitor (A32953, Thermo Scientific) and a phosphatase inhibitor (A32957, Thermo Scientific). As a positive control OR wandering third instar larvae from the same vial were subjected to UV irradiation (10 Gy at 2 Gy/minute, using an Stratagene crosslinker) and allowed to recover for thirty minutes before homogenization in RIPA buffer. Larvae were homogenized in a 500 μL PCR tube with a motorized pestle, mixed with 5X loading buffer (0.5 M Tris-HCl, pH6.8, 5% SDS, 0.0125% bromophenol blue, 12.5% β-mercaptoethanol, 50% glycerol) and 15% per volume β-mercaptoethanol, and boiled for 10 minutes. Protein samples were separated on a 12% polyacrylamide gel and transferred onto PVDF membranes (0.2 μm). Membranes were blocked in a solution of 20% Odyssey Blocking Buffer (LI-COR, 927–50000) in TBS at room temperature for one hour. Membranes were incubated in primary antibodies diluted in TBST at 4˚C overnight. Mouse anti-γH2Av (1,500, UNC93-5.2.1 hybridoma bank) and rabbit anti-H2Av (1,5000, Active Motif Catalog #39715) were used as primary antibodies. Next, membranes were washed four times in TBST buffer, 5 minutes each wash. Membranes were then incubated in secondary antibodies diluted in TBST buffer at room temperature for one hour. Secondary antibodies used in these western blots include IRDye 680RD Goat anti-Rabbit 1:10,000 (LI-COR #92668071) and IRDye 800CW Donkey anti-Mouse 1:10,000 (LI-COR #92632212). Membranes were washed four times in TBST buffer followed by two washes in TBS buffer, 5 minutes each wash. Blots were imaged on an Odyssey scanner. Image files were manipulated in FIJI for analysis and presentation [42].

## Phenotypic analysis

Documentation of $y^2$ and $ct^6$ phenotypes was performed using a stereomicroscope (MZ16 FA; Leica Microsystems) equipped with a CCD color camera (DFC420; Leica Microsystems). A 150 Watt white light source (KL 1500 LCD; Leica Microsystems) set to a color temperature of

3,000 K was used for illumination. Male flies were selected soon after eclosion and aged for five days at 25°C before imaging. All images within each tissue set were collected with the same parameters using Leica Application Suite (Version 2.4.0 R1; Leica Microsystems). Abdomen images were recorded with a gamma correction of 0.5. Image analysis was performed in FIJI [42]. Intensity of the darkest region within the fifth abdominal tergite was measured using a circular ROI with radius of 15 pixels (= 0.045 mm) (S1 Fig). Intensity values from abdomens were inverted before analysis so that darker pigmentation provided a higher score. Regions of abdomens that reflected the light source were excluded from analysis. Wing areas were measured in FIJI [42] using the entire translucent area of the wing (except for the alula, which was sometimes lost in sample preparation) as the ROI.

## Results

### γH2Av colocalizes with *gypsy* insulator components genome-wide

Mutation of *su(Hw)* interferes with DNA damage repair [25] and lack of Su(Hw) protein results in chromosomal aberrations in developing neuroblasts [24]. While these results point to a role in maintaining genome stability, the mechanistic link between Su(Hw) and DNA repair remains uncharacterized. To investigate this possibility, we performed immunostaining of polytene chromosomes from salivary glands of third instar larvae from the Oregon R wild-type stock, using an antibody directed against the phosphorylated form of H2Av (γH2Av), a marker of DNA double strand breaks [60]. This procedure reveals a close association between Su(Hw) and γH2Av throughout each polytene chromosome arm. Close inspection shows γH2Av in nearly all of the Su(Hw) bands (Fig 1A). Linescans of the 2R polytene chromosome show strong covariance in the fluorescent intensity between γH2Av and Su(Hw) (Fig 1A). Likewise, analysis of the immunostaining signal using the entire polytene genome shows a significant colocalization between γH2Av and Su(Hw) (Fig 1A).

An extensive genome-wide association between γH2Av and Su(Hw) is illustrated by the positive PCC values and MCC values above 0.5. Equivalent immunostaining experiments yield similar results for Mod(mdg4)67.2, the isoform of the *mod(mdg4)* locus associated with *gypsy* insulator function (Fig 1B). Bands of Mod(mdg4)67.2 are seen to overlap with bands of γH2Av (Fig 1B) and linescans reveal a close covariance (Fig 1B). The same pattern is recapitulated when examining images and linescans of CP190 (Fig 1C), an insulator protein found at *gypsy* and CTCF insulator sites [14,61]. Quantitative analysis of signal colocalization shows strong positive correlation values when examining genome-wide signal in polytene chromosomes for both Mod(mdg4) and CP190 with γH2Av (Fig 1B and 1C). These findings suggest that γH2Av colocalizes with *gypsy* insulator proteins throughout the genome under normal developmental conditions.

### H2Av is phosphorylated in *gypsy* insulator sites

Previous results demonstrated the colocalization of γH2Av and Su(Hw) insulator components genome-wide. We next asked whether γH2Av is also present at *gypsy* insulator sites by looking at two classical examples of mutations induced by the *gypsy* retrotransposons insertion, $y^2$ and $ct^6$, which are found in the fly stock $y^2$, $w^1$, $ct^6$ (Fig 2). The former is an allele of the *yellow* (*y*) gene in which a *gypsy* insulator site inserted into the region between enhancers for expression in the wing and body of the fly and the promoter, cutting off contact between the enhancers and promoter upstream of the *yellow* gene promoter and resulting in lowered expression of the *yellow* gene and ultimately lighter pigmentation of the adult fly [3,4]. The latter example is an allele of the *cut* (*ct*) gene in which a *gypsy* insulator between the wing margin enhancer and *cut* promoter prevents expression of the *cut* gene in the developing wing. This decreases *cut*

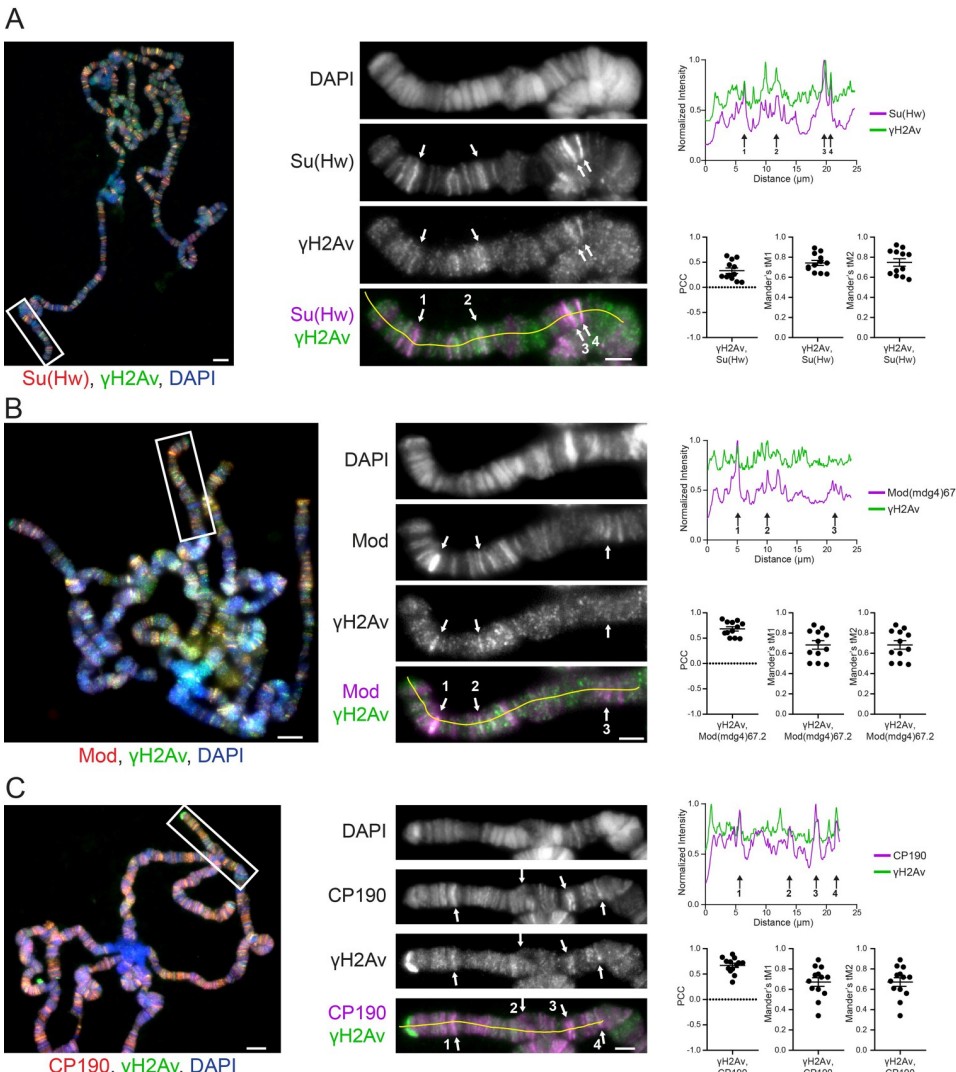

**Fig 1. Insulator proteins colocalize with phosphorylated H2Av in *Drosophila* polytene chromosomes. A**. Colocalization of γH2Av with Su(Hw). **B**. Colocalization of γH2Av with Mod(mdg4)67.2. **C**. Colocalization of γH2Av with CP190. Immunofluorescent micrographs of polytene chromosome squashes obtained from wandering third-instar larvae are shown on the left. Magnified insets are shown in the middle, corresponding to the white boxes in the figures on the left. Scale bars are 5 μm in the figures and 2 μm in the insets. Insets are shown as RGB merge, with DAPI on the blue channel, γH2Av on the green channel, and various insulator proteins on the red channel. Red and green channels are shown independently in grey scale and merged as magenta and green. On the right are linescans corresponding to the yellow lines in the merged insets. Linescan intensities were normalized by dividing each value by the maximum intensity recorded on each channel. Arrows in the images and respective linescans denote regions of strong colocalization. Pearson's Correlation Coefficient (PCC) for γH2Av signal with each insulator protein signal is plotted, with each point representing the polytene genome of each cell. Similarly, Mander's and tM2 are plotted in each cell. Error bars represent one standard error of the mean.

expression in cells of the wing margin, leading to a jagged appearance of the wing margin [12,62]. As it has previously been demonstrated that a functional *gypsy* insulator complex composed of Su(Hw), Mod(mdg4)67.2, and CP190 is required for proper function of the insulator [13,63–66], these sites serve as known examples of genomic loci associated with binding of each of these components.

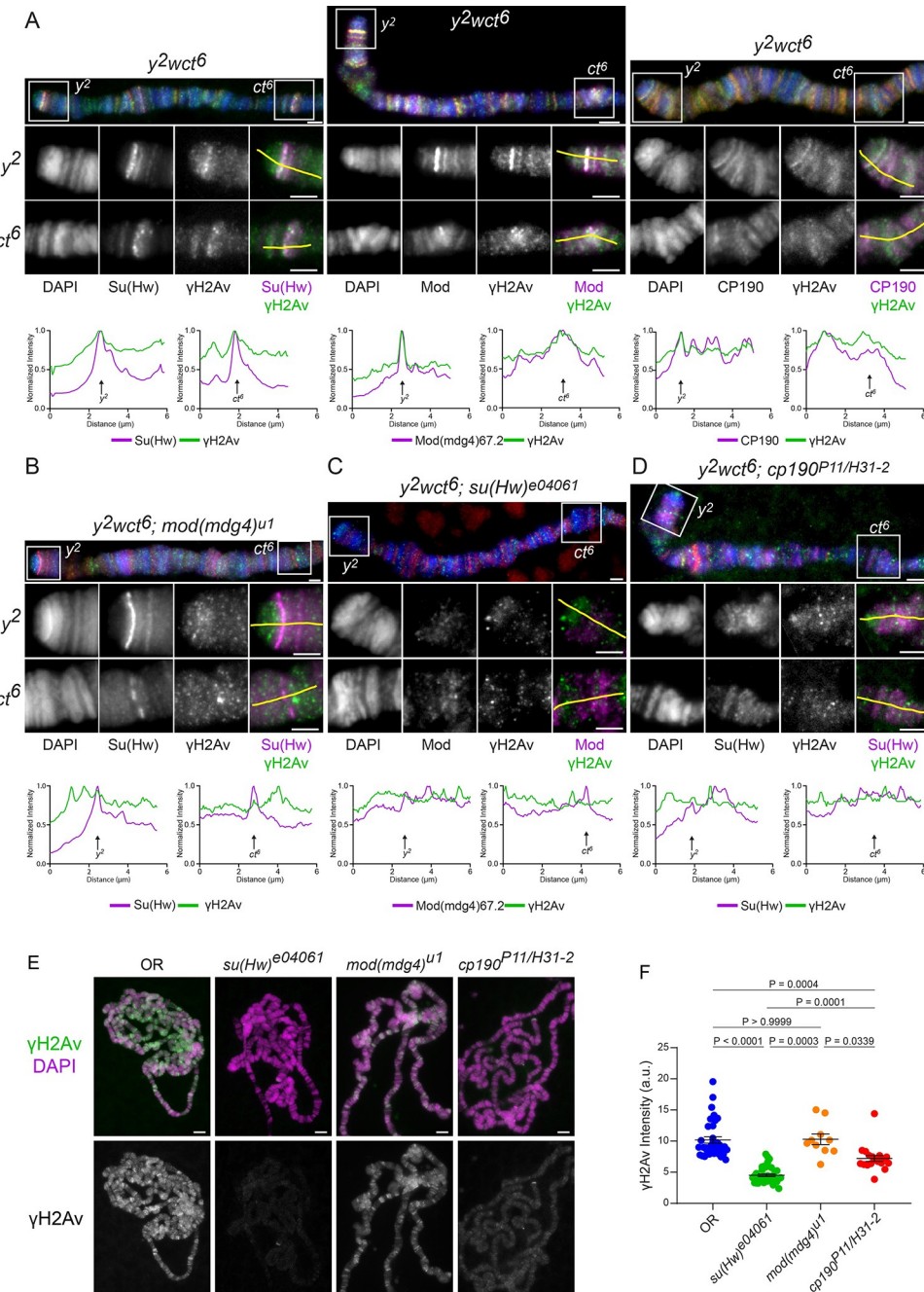

**Fig 2. Phosphorylated H2Av is present at *gypsy* insulator sites and γH2Av levels are significantly reduced in insulator protein mutants. A.** Colocalization between γH2Av and Su(Hw) (left), Mod(mdg4)67.2 (center), and CP190 (right) in a wild type *y²wct⁶* background. **B.** Colocalization between γH2Av and Su(Hw) in a *mod(mdg4)^u1* homozygous background. **C.** Colocalization between γH2Av and Mod(mdg4)67.2 in a *su(Hw)^e04061* homozygous background. **D.** Colocalization between γH2Av and Su(Hw) in a *cp190^P11/H31-2* *trans*-heterozygous background. Immunostaining results are shown of the X polytene chromosome from wandering third instar larvae in the *y²wct⁶* background. Each panel shows an X chromosome, with the *y²* and *ct⁶* sites labelled and delineated with white boxes. Beneath these are insets showing the *y²* and *ct⁶* sites in detail. Insets are shown as RGB merge, with DAPI on the blue channel, γH2Av on the green channel, and various insulator proteins on the red channel. Red and green channels are shown independently in grey scale and merged as magenta and green. Beneath the insets are linescans corresponding to the yellow lines in the merged insets. Arrows represent the approximate sites of each *gypsy* locus. Scale bars are 2 μm in the main figures and insets. Linescan intensities were normalized by dividing each value by the maximum intensity recorded on each channel. **E.** Immunostaining of polytene chromosomes from wandering third instar *Drosophila*

larval salivary glands from various insulator mutant genotypes (listed above each figure). An antibody against phosphorylated H2Av (γH2Av) was used. Scale bars represent 5 μm. **F.** Quantification of fluorescent signals from immunostains with γH2Av shown in E. Each point represents the polytene genome of an individual cell. Error bars represent one standard error of the mean. P-values above the data were determined using an ANOVA performed with Dunnett's T3 multiple comparisons test.

Notably, a strong colocalization was observed for phosphorylated H2Av and each *gypsy* insulator component (Su(Hw), Mod(mdg4)67.2, and CP190) at both $y^2$ and $ct^6$ sites (Fig 2A). Strikingly, this colocalization is lost in $mod(mdg4)^{u1}$ mutants, as seen in chromosomes from $mod(mdg4)^{u1}$ homozygous chromosomes from the $y^2w^1ct^6$; $mod(mdg4)^{u1}$/TM6B, $Tb^1$ fly stock (Fig 2B). Su(Hw) is still recruited to *gypsy* insulator sites in $mod(mdg4)^{u1}$ as expected based on previous reports of *mod(mdg4)* mutants [13,67]; more significant, however, is the observation that γH2Av is no longer observed colocalizing with Su(Hw) at $y^2$ or $ct^6$. Likewise, γH2Av is no longer observed at either $y^2$ or $ct^6$ in the $su(Hw)^{e04061}$ homozygous mutant background from the fly stock $w^{1118}$; PBac(RB)$su(Hw)^{e04061}$/TM6B, $Tb^1$ (Fig 2C). The lack of Mod(mdg4)67.2 at $y^2$ and $ct^6$ in the absence of Su(Hw) agrees with previous reports, which implicate Su(Hw) as necessary for recruitment of Mod(mdg4)67.2 to *gypsy* loci [13,68], while the lack of γH2Av implies that either Su(Hw) is directly needed to maintain H2Av in a phosphorylated state or that a complete *gypsy* complex containing Mod(mdg4)67.2 is required. The *trans*-heterozygous $cp190^{P11/H31-2}$ mutant chromosomes generated using fly stock $y^2w^1ct^6$; $cp190^{H31-2}$/TM6B, $Tb^1$ crossed with $y^2w^1ct^6$; $cp190^{P11}$/TM6B, $Tb^1$ (Fig 2D) showed a decrease in the amount of Su(Hw) present at $y^2$ and $ct^6$, consistent with a previous report that found reductions in both Su(Hw) and Mod(mdg4)67.2 in the polytene chromatin of *cp190* mutants, [68]. Mutation of *cp190* also significantly reduced levels of γH2Av at the two *gypsy* loci examined. All together, these results indicate that reduction of any of the canonical *gypsy* insulator components is sufficient to strongly reduce the levels of γH2Av at these sites.

### *Gypsy* insulator proteins are required for normal chromosomal distribution of phosphorylated H2Av

To further our understanding of the relationship between γH2Av and insulator complexes we asked whether or not mutation of genes coding for *gypsy* insulator complex members would affect γH2Av levels. Immunostaining of polytene chromatin revealed an almost complete elimination of γH2Av from chromosomes of $su(Hw)^{e04061}$ mutants from fly stock $w^{1118}$; PBac(RB)$su(Hw)^{e04061}$/TM6B, $Tb^1$ (Fig 2E and 2F). These mutants lack *su(Hw)* expression due to the insertion of a *piggy-bac* transposon element in the gene (Baxley et al., 2011). Mutation of *cp190* using two effective null alleles (from fly stocks $y^2w^1ct^6$; $cp190^{H31-2}$/TM6B, $Tb^1$ and $y^2w^1ct^6$; $cp190^{P11}$/TM6B, $Tb^1$) also resulted in less γH2Av signal in the immunostained chromosomes, however, the reduction was not as severe as was seen in the $su(Hw)^{e04061}$ mutant (Fig 2E and 2F). In contrast with *su(Hw)* and *cp190*, mutation of *mod(mdg4)* had no significant effect on the amount of γH2Av in chromatin (Fig 2E and 2F).

Notably, the colocalization between Mod(mdg4)67.2 and γH2Av was strongly reduced in the $su(Hw)^{e04061}$ background (Fig 3A and 3F). This is not surprising given that γH2Av signal is significantly reduced in $su(Hw)^{e04061}$ (Fig 2E and 2F) and that Mod(mdg4)67.2 does not bind at Su(Hw) sites in the absence of Su(Hw) [13]. Linescans of polytene chromosomes and quantitative colocalization analysis show no correlation between the signals from Mod(mdg4)67.2 and γH2Av in $su(Hw)^{e04061}$ mutant polytene chromosomes (Fig 3A). Similar results were obtained for colocalization between CP190 and γH2Av in the $su(Hw)^{e04061}$ mutant (Fig 3B).

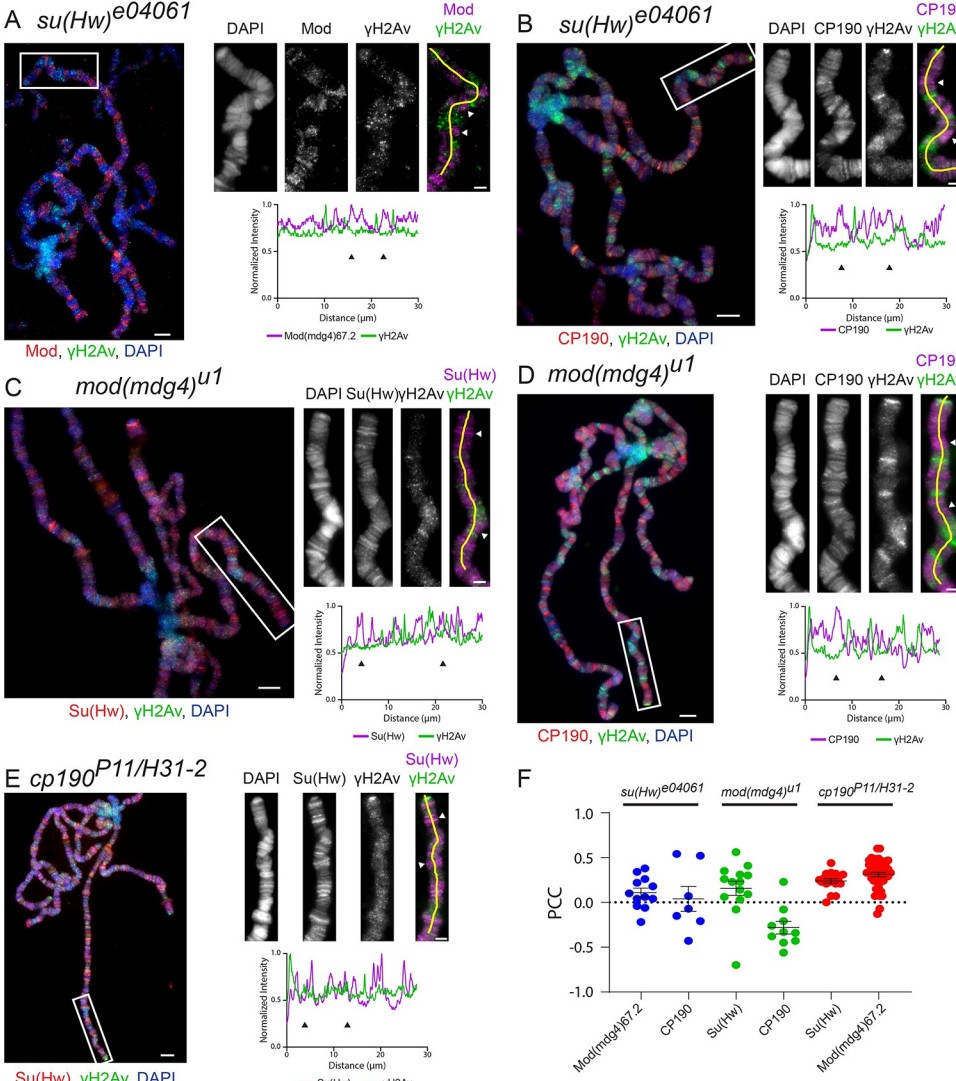

**Fig 3. Insulator components are interdependent for their colocalization with γH2Av in *Drosophila* polytene chromosomes. A.** Colocalization of γH2Av with Mod(mdg4)67.2 in *su(Hw)^e04061^*. **B.** Colocalization of γH2Av with CP190 in *su(Hw)^e04061^*. **C.** Colocalization of γH2Av with Su(Hw) in *mod(mdg4)^u1^*. **D.** Colocalization of γH2Av with CP190 in *mod(mdg4)^u1^*. **E.** Colocalization of γH2Av with Su(Hw) in *cp190^P11/H31-2^*. **F.** Pearson's Correlation Coefficient (PCC) for γH2Av signal with each insulator signal is plotted, with each point representing the polytene genome of each cell. Error bars represent one standard error of the mean. PCC values are grouped by genotype (red = *su(Hw)^e04061^*, green = *mod(mdg4)^u1^*, blue = *cp190^P11/H31-2^*). Immunostains of polytene chromosomes are shown on the left. Magnified insets are shown to the right of each figure, corresponding to the white boxes in the figures on the left. Scale bars are 5 μm in the figures and 2 μm in the insets. Beneath the insets are linescans corresponding to the yellow lines in the merged insets. Linescan intensities were normalized by dividing each value by the maximum intensity recorded on each channel. Arrowheads denote regions enriched for insulator proteins but not γH2Av.

Likewise, immunostaining for Su(Hw) in the loss of function *mod(mdg4)^u1^* mutant (from the *y^2^w^1^ct^6^; mod(mdg4)^u1^/TM6B, Tb^1^* stock) revealed significantly less colocalization as seen by visual inspection, linescans, and quantitative analysis (Fig 3C and 3F). Linescans from immunostains of the *mod(mdg4)^u1^* mutant with either Su(Hw) (Fig 3C) or CP190 (Fig 3D) along with γH2Av demonstrate a further disruption of the interaction between γH2Av and insulator complexes. The lack of colocalization between γH2Av and Su(Hw) or CP190 is

reflected in decreased PCC values (Fig 3F), with negative values for CP190 indicating anticorrelation between the CP190 and γH2Av signals. Su(Hw) was also found to colocalize less with γH2Av in the null *cp190^P11/H31-2^* mutant background (from fly stocks *y²w¹ct⁶; cp190^H31-2^/ TM6B, Tb¹* and *y²w¹ct⁶; cp190^P11^/TM6B, Tb¹*) through the area of chromosome 2R examined (Fig 3E) and genome wide (Fig 3F). Taken together, these results indicate that stable accumulation of γH2Av in Su(Hw) insulator sites requires having the entire insulator complex intact.

## γH2Av and Su(Hw) are enriched at TAD boundaries

To determine if the patterns we observe in polytene chromosomes are conserved in other cell types, we used publicly available ChIP data for γH2Av and Su(Hw) from Kc167 cells (SRX1299942 SRA) [49]. We used SeqMiner to compare the two ChIP-seq datasets by heatmap analysis. The peak summits for γH2Av were taken as the reference coordinate for a heatmap and profile analysis comparing Su(Hw) and γH2Av distributions (Fig 4A). The profile of mean read densities for both proteins also shows a significant overlap between Su(Hw) and γH2Av peaks at γH2Av peaks flanked by 10,000 bp (Fig 4B). These results support our observation that the genomic distribution of Su(Hw) and γH2Av overlap in polytene chromosomes.

Because insulator proteins are frequently associated with the boundaries of Topologically Associating Domains (TADs), we next asked whether γH2Av is also enriched at TAD boundaries. To address this question, we used publicly available data and obtained a map of all *Drosophila* TADs as determined by Hi-C using *Drosophila* Kc167 cells [59]. Heatmap analysis shows a strong association between the density distribution of H2Av nucleosomes and TAD boundaries (Fig 4C). Generally, we found homotypic H2Av nucleosomes are enriched at the boundaries, whereas heterotypic nucleosomes have a significant density drop at TAD boundaries.

Enrichment in γH2Av and Su(Hw) is also observed at the TAD boundaries (Fig 4C). Signal enrichment can be grouped in two major clusters, where the most significant difference is the relative enrichment of Su(Hw) and γH2Av at the boundaries (Fig 4D and 4E). In cluster 2, heterotypic H2Av and γH2Av have similar enrichment levels through the DNA flanking the boundary. At the center of the boundary, however, is the homotypic instead of the heterotypic H2Av that has an enrichment similar to that of γH2Av, with both intensity profiles (H2Av Homotypic and γH2Av) significantly more elevated than in the flanking DNA (Fig 4E). Interestingly, Cluster 1 has the opposite pattern. In cluster 1, homotypic H2Av and γH2Av have similar enrichment densities through the DNA flanking the boundary and remain elevated at the boundary center. However, γH2Av and Su(Hw) intensities are similar and higher than that of the homotypic and heterotypic H2Av nucleosomes (Fig 4D). Cluster 3 seems to be characterized by a higher enrichment of Su(Hw) in the flanking regions (see S1 Fig for specific examples).

We found that the association of γH2Av with TAD boundaries is very similar to that of Su(Hw) and other insulator proteins. One example of this is a TAD boundary that flanks a TAD containing the homeotic gene *Abdominal A* (*Abd-A*). Like other boundaries associated with the homeobox gene cluster [1,69], this boundary is enriched in the insulator proteins Su (Hw), Modifier of Mdg4, CP190, CTCF, and HIPP1. Here, we show this boundary is equally enriched in γH2Av (Fig 4F). Moreover, when observing the distribution of H2Av nucleosomes at this site, it appears that the γH2Av peak does not colocalize with a nucleosome. In another example, two boundaries flank a TAD that contains the developmentally regulated pair rule gene *eve* (Fig 4G). Each boundary is enriched in γH2Av; however, in the left boundary the γH2Av peak overlaps at least partially with the nucleosomal H2Av, whereas in the right boundary the γH2Av peak does not colocalize with nucleosomes either. This particular

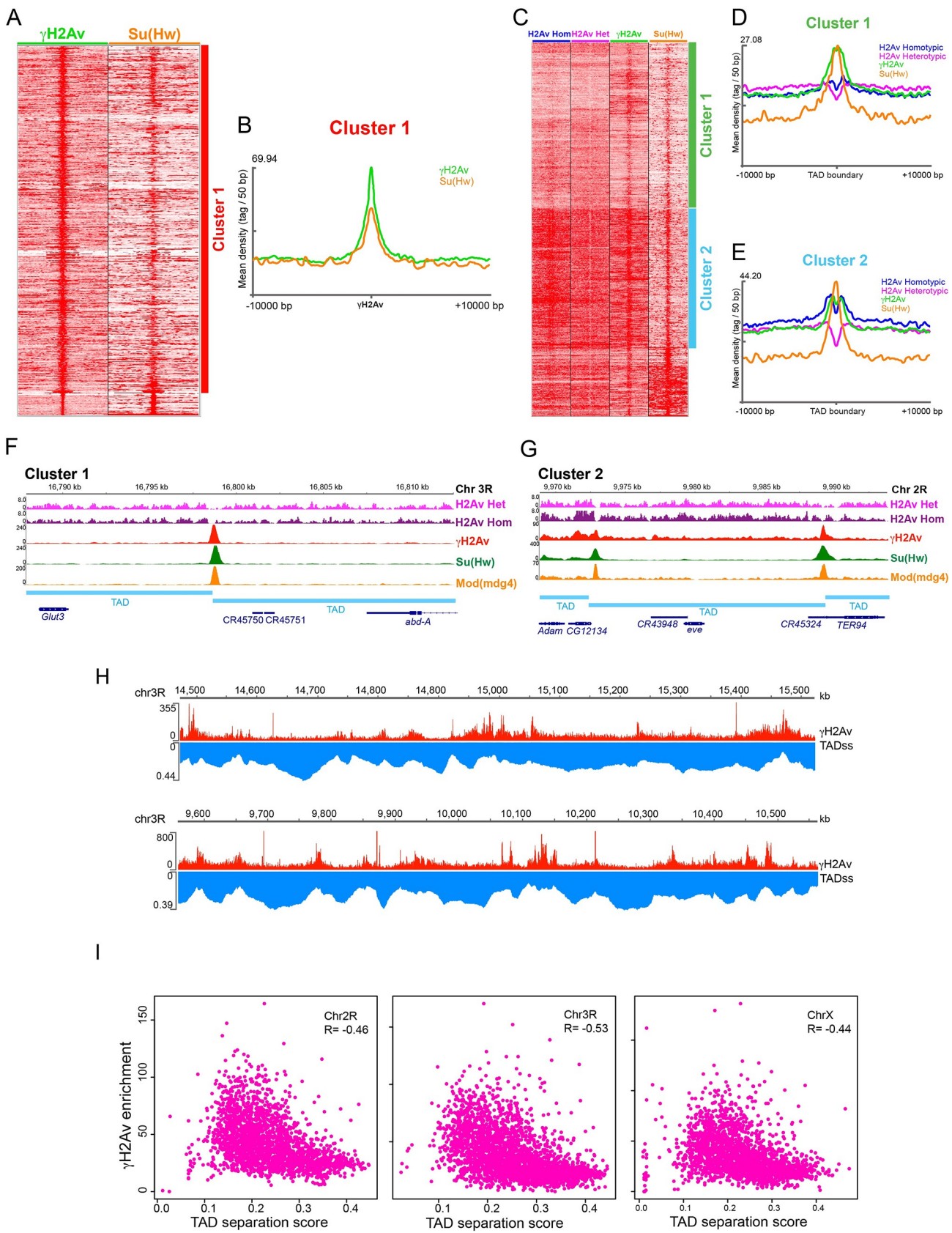

**Fig 4. Phosphorylated H2Av associates with TAD boundaries. A.** Heatmap comparing the intensity distributions of γH2Av and Su(Hw) using γH2Av peaks as reference. **B.** Mean read density profiles of γH2Av and Su(Hw) centered at γH2Av peaks. **C.** Heatmap comparing the intensity distributions of H2Av nucleosomes, γH2Av, and Su(Hw) using TAD boundaries as a reference. **D.** and **E.** Mean read density profiles of two major clusters from heatmap in C, centered at TAD boundaries. **F.** Peak profile of insulator proteins, including γH2Av and the H2Av nucleosome distribution at the left boundary of the *Abd-A* TAD. **G.** Peak profile of insulator proteins, including γH2Av and the H2Av nucleosome distribution at the Homie insulator flanking the pair rule gene *eve*. (From *A* through *G* dm6 was used as reference genome). **H.** Enrichment of γH2Av compared to TAD-separation scores in two chromosome 3R segments. **I:** Scatterplots and Spearman correlation coefficients comparing γH2Av enrichment with TAD-separation scores in *Drosophila* chromosomes X, 2R and 3R (in H and I, dm3 was used as a reference genome).

boundary corresponds to the well characterized Homie insulator [70]. Further experimental evidence is necessary to confirm the possibility that non-nucleosomal γH2Av may be present at some Su(Hw) insulator sites. Additional specific examples illustrating the presence of γH2Av at TAD boundaries are presented in S1 Fig.

Given these observations, and because H2Av is broadly distributed through the genome, we decided to use a quantitative approach to further ask whether phosphorylated H2Av is enriched at TAD boundaries. To this end, we compared the coverage of γH2Av ChIP-seq reads with the distribution of the TAD-separation scores along *Drosophila* chromosomes. TAD-separation scores depict a quantitative measure of the frequency of contacts between flanking DNA regions and was developed by Ramirez et al. [59] by scoring the Hi-C contact read coverage along the chromosome. A low separation score at a specific site in the chromosome means a low number of contact reads mapping to that site, indicating a low frequency of contacts between the adjacent DNA sequences. The lowest separation scores mostly correspond with boundaries between TADs. Fig 4H shows two 1 Mb regions from chromosome 3R, along with the γH2Av coverage and the TAD-separation score distributions. Higher enrichment in γH2Av is observed at sites of low TAD-separation scores and a depletion of γH2Av can be seen in areas of higher scores.

To compute the coverage for both variables along entire chromosomes we used DeepTools multibigwigSummary (Galaxy Version 3.5.1.0.0). This tool generates an output file with the number of read counts, for both variables (γH2Av ChIP-seq reads and TAD separation sores), in 10,000 bp bins along the entire chromosome. We used these files to produce scatterplots and to calculate the Spearman correlation between both distributions, γH2Av and TAD-separation scores for chromosomes X, 3R and 2R (Fig 4I). Results show a negative correlation between the coverage of γH2Av and the TAD-separation scores, indicating that there is consistently a higher γH2Av enrichment at sites where the separation scores are lower. Because low separation scores are associated with TAD boundaries, this result supports our observation that there is an association between γH2Av with TAD boundaries genome wide.

## Phosphorylated histone H2Av is found at insulator bodies

In diploid *Drosophila* cells, insulator proteins have been shown to aggregate in the nucleus forming numerous foci that have been designated as insulator bodies [16]. The exact purpose of these bodies remains unknown, but recent reports have shed new light on their formation and dynamics. In a previous work, our lab demonstrated that insulator bodies form after dissociation of insulator proteins from chromatin and that insulator body formation can be induced under conditions of high osmotic pressure in both diploid cells and polytene salivary gland cells [20]. Here, our results show localization of γH2Av to insulator bodies using immunostaining of S2 cells and salivary gland cells (OR) for γH2Av and Su(Hw) in hypertonic media (Fig 5A–5E). Quantitative colocalization analysis shows a clear and significant increase in colocalization between Su(Hw) and γH2Av after hypertonic stress, given the presence of γH2Av in insulator bodies (Fig 5D). The absence of a significant number of insulator bodies in

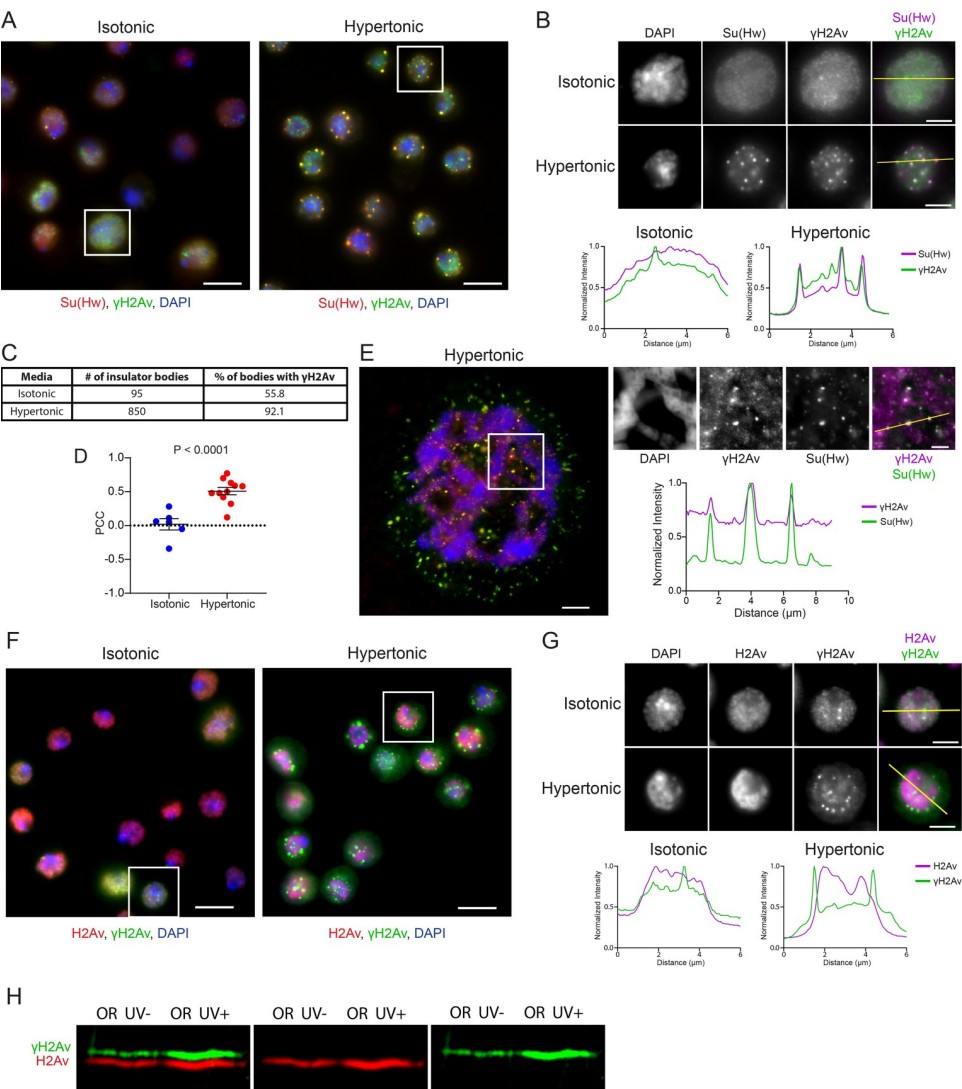

**Fig 5. Phosphorylated H2Av is a component of insulator bodies. A.** Immunostaining of *Drosophila* S2 cells in isotonic media (left) and hypertonic media (right). Insulator bodies formed during osmotic stress are labelled with Su(Hw) (red). Phosphorylated H2Av (green) colocalizes with Su(Hw) in insulator bodies. Scale bars are 5 μm and insets for analysis are delineated by white boxes. **B.** Magnified view of insets showing insulator body formation in hypertonic, but not isotonic, conditions. Insets are shown as RGB merge, with DAPI on the blue channel, γH2Av on the green channel, and Su(Hw) on the red channel. Beneath the insets are linescans corresponding to the yellow lines in the merged insets. Scale bars represent 2 μm. **C.** Table showing the ratio of insulator bodies with significant amounts of γH2Av. **D.** Pearson's Correlation Coefficient (PCC) for γH2Av signal with Su(Hw) signal is plotted for isotonic versus hypertonic conditions, with each point representing a field of S2 cells. Error bars represent one standard error of the mean. The P-value was determined using an unpaired two-tailed Student's T-test. **E.** Polytene chromosomes under hypertonic conditions. Colocalization between γH2Av and Su(Hw) is shown in the insets and linescan. The scale bar in the wide view represents 5 μm, the scale bar in the inset represents 2 μm. **F.** Immunostaining of *Drosophila* S2 cells in isotonic media (left) and hypertonic media (right). Phosphorylated H2Av (green) localizes to insulator bodies while unphosphorylated H2Av (red) does not. Scale bars are 5 μm and insets for analysis are delineated by white boxes. **G.** Magnified insets from F. Beneath the insets are linescans corresponding to the yellow lines in the merged insets. Scale bars represent 2 μm. Linescan intensities in B, E, and G were normalized by dividing each value by the maximum intensity recorded on each channel. **H**. Western blot showing expression of γH2Av and H2Av in UV-light treated and untreated OR larvae demonstrating specificity of anti-γH2Av and anti-H2Av antibodies used in F.

unstressed cells and the lack of optical resolution inherent to diploid nuclei prevents a meaningful analysis of colocalization between the two signals in isotonic media (Fig 5C and 5D).

Next, we asked whether γH2Av is required for colocalization of H2Av to insulator bodies. First, we determined the specificity of each antibody, mouse anti-γH2Av (hybridoma bank) and rabbit anti-H2Av (Active Motif) against the phosphorylated and unphosphorylated versions of H2Av, respectively. To confirm that each antibody specifically recognizes its protein target, we performed a western-blot simultaneously proving γH2Av and unphosphorylated H2Av. We performed PAGE with proteins extracted from OR larvae (UV-light treated and untreated). The western-blot was incubated with anti-H2Av and anti-γH2AV primary antibodies, and with fluorescently labeled secondary antibodies (Odyssey). Results show that both UV-light treated and non-treated larvae express γH2Av, although as expected UV-treated larvae express γH2Av more abundantly. This result further supports the immunostaining and ChIP data presented here showing that γH2Av is normally expressed in cells under physiological conditions. Because the phosphorylation in γH2AV affects the migration of γH2Av in the polyacrylamide gel, this result also shows that anti-γH2Av and anti-H2Av antibodies are highly specific and do not cross-react (Fig 5H). To test whether γH2Av is required for translocation of H2Av to insulator bodies, we performed immunostaining using the same mouse anti-γH2Av and rabbit anti-H2Av antibodies in S2 cells. Results show there is no accumulation of unphosphorylated H2Av in insulator bodies (Fig 5F and 5G), suggesting that phosphorylation of this histone variant is required for its localization to insulator bodies.

## Phosphatase inhibition stabilizes interactions between γH2Av and *gypsy* insulator proteins

In order to determine how H2Av phosphorylation affects insulator complex dynamics, we used okadaic acid (OA), a potent inhibitor of serine/threonine phosphatases PP1 and PP2A *in vitro* and *in vivo* [71,72]. To limit the inhibitory effect of okadaic acid to PP2A phosphatase and avoid affecting the function of PP1, we used a concentration of 50 nM OA [73]. As expected, results show an increase in the amount of phosphorylated H2Av bound to the polytene chromatin in the presence of okadaic acid compared to the untreated control salivary glands chromosomes from an OR fly stock (S2B Fig). This suggests that the okadaic acid is inhibiting PP2A from dephosphorylating γH2Av in chromatin. Quantification of the antibody signals also indicates a significant increase in the amount of Su(Hw) signal in the presence of okadaic acid compared to untreated OR polytene chromosome samples (S2B Fig). Interestingly, examination of the polytene chromosomes showed no significant effect of okadaic acid on the distribution of insulator proteins or their colocalization with phosphorylated H2Av (S2A, S2C and S2D Fig).

Intriguingly, immunostaining of salivary glands from insulator protein mutants after incubation in okadaic acid revealed a significant rescue of both γH2Av levels and its colocalization with components of the *gypsy* insulator complex. For example, γH2Av signals are not detected or are very week in *su(Hw)* and *cp190* mutants, respectively (Fig 3), but they are relatively strong in *su(Hw)* and *cp190* mutants (stocks *w^1118^*; PBac(RB)*su(Hw)^e04061^*/TM6B, *Tb^1^* and trans-heterozygous cross between lines *y^2^w^1^ct^6^*; *cp190^H31-2^*/TM6B, *Tb^1^* and *y^2^w^1^ct^6^*; *cp190^P11^*/ *TM6B, Tb^1^*, respectively) in okadaic acid-treated samples (S3 Fig). This result suggests that in *su(Hw)* mutants, Mod(mdg4)67.2 weakly interacts with CP190 and that this interaction is enhanced by γH2Av (S3A and S3B Fig). Similar results were obtained when staining for Su(Hw) and CP190 in the *mod(mdg4)^u1^* mutant (from the *y^2^w^1^ct^6^*; *mod(mdg4)^u1^*/TM6B, *Tb^1^* stock), with both proteins showing colocalization with phosphorylated H2Av only after incubation with okadaic acid (S3C, S3D and S3F Fig). As with the *su(Hw)^e04061^* mutant, this is in

contrast to the untreated mutant samples in which γH2Av and *gypsy* insulator complexes do not colocalize. Staining of the *trans*-heterozygous *cp190^{P11/H31-2}* mutant for Su(Hw) after okadaic acid treatment yielded the same response, with colocalization between Su(Hw) and γH2Av being rescued compared to the untreated mutant (S3E and S3F Fig). These findings point to a model in which γH2Av acts as a component of *gypsy* insulator protein complexes. Phosphorylated H2Av may be required for *gypsy* insulator complex formation or stabilization, and may play an essential role in insulator function.

We next asked if this role of γH2Av in *gypsy* insulator dynamics is limited to chromatin-bound insulators or if it extends to other cellular functions of insulators. To this end, S2 cells were exposed to osmotic stress to induce insulator body formation with the goal of determining if γH2Av is required for insulator body formation or recovery after stress (Fig 6A). As a control to ensure okadaic acid alone does not induce insulator body formation, cells were incubated in isotonic media with okadaic acid. These cells showed similarly low numbers of insulator bodies per cell as cells in untreated isotonic media (Fig 6B). Osmotic stress was introduced by increasing the salt concentration as shown in the methods section [20], resulting in the formation of insulator bodies (Fig 6A and 6C). No significant difference was seen in the ratio of cells that contained insulator bodies when comparing okadaic acid-treated and -untreated samples (Fig 6B). Of particular interest, however, is the finding that after the osmotic stress media is replaced with isotonic media, cells treated with okadaic acid recover significantly less than cells not exposed to okadaic acid, retaining a greater number of insulator

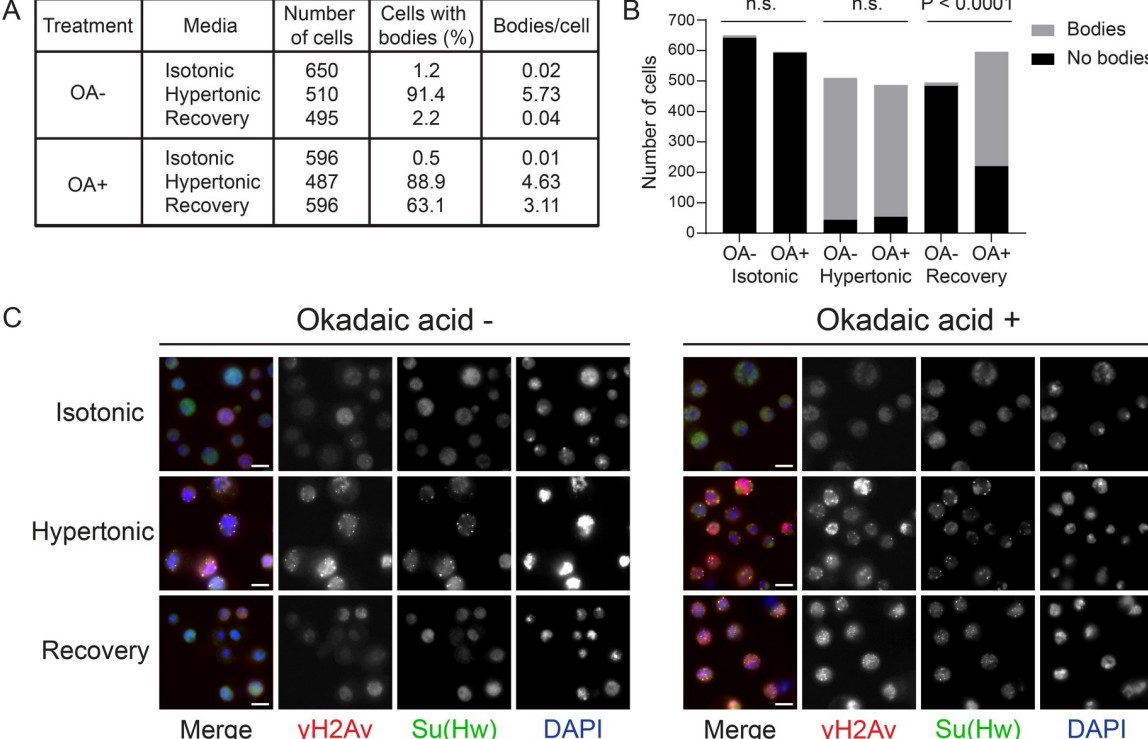

**Fig 6. Phosphatase inhibition prevents recovery from insulator body formation after osmotic stress. A.** Tabulated results from OA (okadaic acid) treatment in osmotic stress and recovery. **B.** Bar graph showing the number of cells displaying bodies (grey) and those not displaying bodies (black). OA- and OA+ treatments are shown for each osmotic condition. P-values were determined using Fisher's exact test. n. s. = not significant. **C.** Osmotic stress and recovery in the absence of okadaic acid (left) in the presence of okadaic acid (right). Representative images are shown of S2 cells in isotonic media, in hypertonic stress media, and recovering in isotonic media. Merged images are shown on the left, with each channel shown independently in greyscale. Scale bars represent 5 μm.

bodies throughout recovery (Fig 6B and 6C). These results put into context our finding that γH2Av is present in insulator bodies (Fig 5) and imply that phosphorylated H2Av must be maintained in the insulator body as part of the normal osmotic response. Preventing dephosphorylation of γH2Av by phosphatase inhibition counteracts resolution of insulator bodies during isotonic recovery, suggesting this is an essential part of the mechanism governing the cellular response to osmotic stress.

## H2Av contributes to *gypsy* insulator function

Our earlier results strongly suggest a correlation between γH2Av and *gypsy* insulator components, including colocalization on polytene chromatin (Figs 1 and 2) and in insulator bodies (Fig 5). Based on the correlations described above between *gypsy* insulator proteins and γH2Av, we wondered whether or not mutation of *His2Av*, the sole H2A variant gene found in the *Drosophila* genome [74], would affect *gypsy* insulator function. As the null mutation of *His2Av* (*His2Av$^{810}$*) is homozygous lethal at the third instar larval stage [75], we were precluded from observing how a complete lack of H2Av would affect the *yellow* and *cut* phenotypes seen in adults. We therefore set up a series of crosses to examine the phenotype of adult animals that carried heterozygous mutations for both *His2Av* and *su(Hw)* using the following stocks: $y^2w^1ct^6$; *His2Av$^{810}$*/TM6B; $Tb^1$, $y^2w^1ct^6$; *PBac(RB)su(Hw)$^{e04061}$*/TM6B, $Tb^1$and $y^2w^1ct^6$; $w^{1118}$; *su(Hw)$^V$*/TM6B, $Tb^1$ (to generate trans-heterozygous *su(Hw)$^{e04061}$* over *su(Hw)$^V$*).

The effects of these mutants on the $ct^6$ allele were measured by examining the wing margins. Such effects are often reported with subjective scoring based on the examiner's interpretation. For a more objective and reproducible method, wing width was measured as a proxy for *cut* gene activity, as wings with the $ct^6$ phenotype are less wide due to the decreased cell proliferation associated with this phenotype. Width of the wing is calculated as the area divided by the feret diameter (a measure of length). Notably, flies that contained heterozygous mutations for both *His2Av$^{810}$* and *su(Hw)$^{e04061}$* displayed wider wings, while both single heterozygotes showed varying degrees of cuts in the margin (Fig 7B). To verify this finding, the null *su(Hw)$^V$* allele was next tested in conjunction with *His2Av$^{810}$*. Consistent with the *His2Av$^{810}$*/su(Hw)$^{e04061}$ genotype, *His2Av$^{810}$*/su(Hw)$^V$ double heterozygotes had significantly wider wings than either single heterozygote (Fig 7B). This increased suppression of *gypsy* insulator phenotypes in *su(Hw)* heterozygous backgrounds implies that H2Av is involved in *gypsy* insulator function.

We also tested the effect of *His2Av* mutation on $y^2$, another *gypsy* insulator phenotype. Effects of these mutations on the $y^2$ allele were tested by measuring the degree of pigmentation in the wings and in the darkened A5 tergite found in male flies. Heterozygous *HisAv$^{810}$* or *su(Hw)$^{e04061}$* mutations by themselves have little effect on expression of *yellow* in the abdomen (Fig 7C). In contrast, abdomens in the double heterozygous *His2Av$^{810}$*/su(Hw)$^{e04061}$ mutants were significantly darker than the single heterozygous mutants (Fig 7C). This implies a reduction in the enhancer-blocking capacity of the *gypsy* insertion upstream of *yellow* and suggests a functional role for H2Av in *gypsy* insulator function. In order to exclude the possibility that second-site mutations in the *su(Hw)$^{e04061}$* background were influencing this rescue, we crossed the null *su(Hw)$^V$* allele with the *His2Av$^{810}$* mutant. Flies doubly heterozygous for *su(Hw)$^V$* and *His2Av$^{810}$* showed a statistically significant increase in pigmentation compared to the *His2Av$^{810}$* heterozygote, but not the single *su(Hw)$^V$* mutant (Fig 7C). This discrepancy between *su(Hw)$^{e04061}$* and *su(Hw)$^V$* may be due to mutation of the neighboring *RpII15* gene (an RNA Pol II subunit) in the *su(Hw)$^V$* chromosome [76], which may have an epistatic effect on this phenotype.

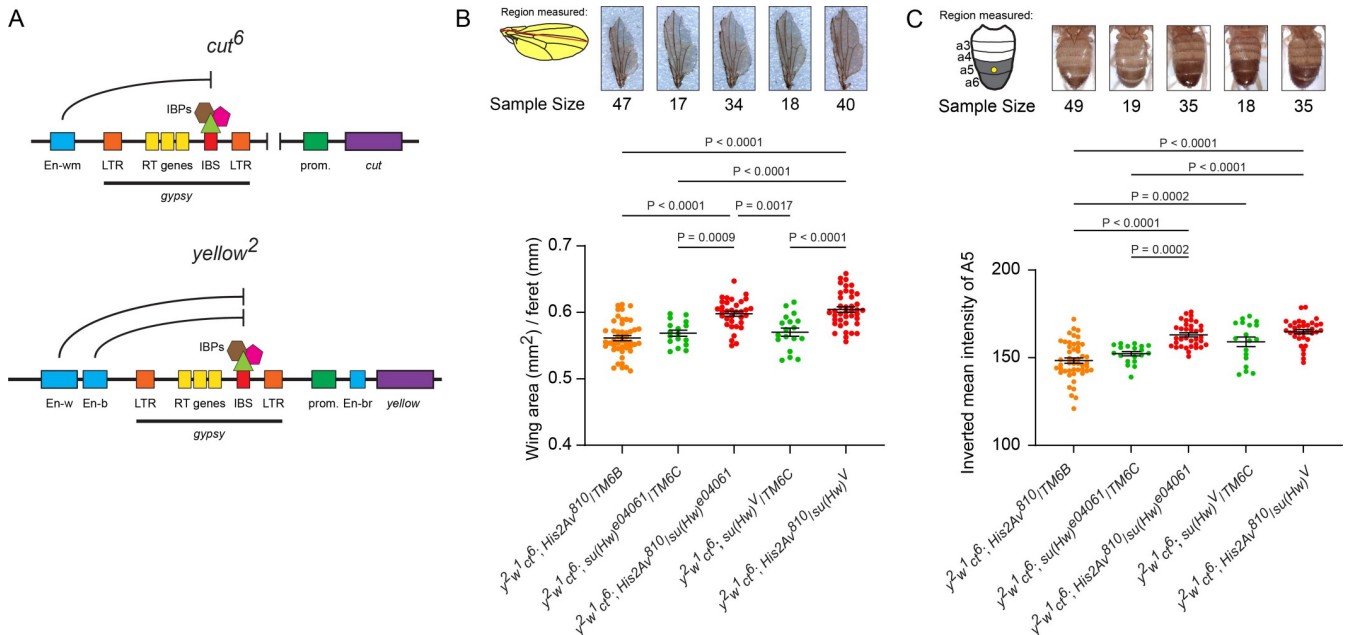

**Fig 7. H2Av contributes to *gypsy* insulator function. A.** Illustration of the upstream regulatory elements found in the $ct^6$ and $y^2$ alleles. Gene coding sequences (purple) are regulated by coordinated contacts between enhancers (blue) and promoters (green). In each case, the *gypsy* retrotransposon (denoted with the thick black bar) has been inserted between a promoter and at least one enhancer. LTR = long terminal repeats, RT genes = retrotransposon genes (*gag*, *pol*, and *env*), IBS = insulator binding site, IBPs = insulator binding proteins (Su(Hw), Mod(mdg4)67.2, and CP190). When the IBS is bound by a complete insulator complex, it interrupts communication between the promoter and distal enhancers. In $cut^6$ this prevents expression of *cut* by the wing margin enhancer (En-wm), while in $yellow^2$ the insulator prevents expression of *yellow* directed by the wing enhancer (En-w) and the body enhancer (En-b) but not the bristle enhancer (En-br) between the *yellow* promoter and transcription start site. Genomic distances are not drawn to scale. **B.** Wing area in mm$^2$ (yellow shaded area) was divided by feret diameter (red line). Examples of wings are displayed above their respective genotypes. **C.** $y^2$ phenotype scoring in the male abdomen. The illustration on the left shows the male abdomen with abdominal segments 3–6 labelled. A circular ROI (yellow circle) was sampled from the a5 tergite of each male to determine the degree of pigmentation. The graph depicts the mean pixel intensity from individual flies by genotype. Example pictures of abdomens are displayed above their respective genotypes. In **B** and **C**, green dots represent *su(Hw)* heterozygotes arising from each cross, orange dots represent *His2Av$^{810}$* heterozygotes, and red dots indicate flies doubly heterozygous for *His2Av$^{810}$* and *su(Hw)*. Above each data set is the sample size P-values from an ordinary one-way ANOVA performed with Tukey's multiple comparisons test, with significant differences at P ≤ 0.05. P-values greater than 0.05 are not shown.

## Discussion

Our results show that γH2Av colocalizes with Su(Hw) insulator complexes throughout the genome, and that both are enriched at TAD boundaries. Using mutations in genes encoding insulator proteins we show that disruption of the insulator complex prevents stable phosphorylation of H2Av. Phosphorylated H2Av levels are regained in insulator mutants after phosphatase inhibition. We provide evidence that this histone variant is involved in insulator activity, as dephosphorylation of γH2Av is necessary for dissolution of insulator bodies and reducing the genetic dose of H2Av in a sensitized *su(Hw)* heterozygous mutant background partially rescues *gypsy* insulator phenotypes.

Chromatin insulator proteins were initially characterized by their enhancer-blocking properties and their ability to prevent the spread of heterochromatin, and more recently by their role in large-scale genome organization [3–5,9]. In addition to these canonical properties, our lab has uncovered roles of insulators in other aspects of cell metabolism including the osmotic stress response [20] and genome stability [77]. It is now established that mutation of CTCF, the only insulator protein found in humans, predisposes cells to cancer formation [78–80] through increased rates of unrepaired DNA damage. Here, we report a functional relationship between a *Drosophila* insulator and H2Av, the sole histone H2A variant in fruit flies. H2Av in

*Drosophila* performs functions associated with both the mammalian histone variants H2AX and H2AZ [34]. Like H2AX, H2Av is phosphorylated in response to DNA double strand breaks (DSBs) and serves as a chromosomal mark to recruit DNA repair proteins [60,81].

Our initial experiments in *Drosophila* polytene chromosomes revealed a striking correlation between the binding sites of phosphorylated H2Av and insulator proteins at *gypsy* insulator sites (Fig 2) and at Su(Hw) sites elsewhere in the genome (Figs 1 and 3), supporting the notion that Su(Hw) is involved in maintaining genome integrity [24,25]. Importantly, this colocalization depends on having a complete and stable Su(Hw) insulator complex, as mutating any of the three core insulator components reduces the levels and the coincidence of γH2Av signals with the remaining insulator proteins (Fig 3). Some insight into the mechanism behind this phenomenon comes from experimental inhibition of PP2A, the phosphatase responsible for dephosphorylating γH2Av after resolution of double strand breaks [82]. We found that the amount of γH2Av in polytene chromosomes increases after PP2A inhibition. Surprisingly, the amount of Su(Hw) bound to the polytene chromosomes also significantly rose when tissues were treated with the PP2A inhibitor (S2 Fig). Indeed, we show that inhibition of PP2A rescues *gypsy* insulator complex formation in tissues missing one of the three core insulator binding proteins (S3 Fig). This supports the notion that Su(Hw) insulators are stabilized by the presence of phosphorylated H2Av.

We observed strong reductions of γH2Av signal in *su(Hw)*$^{e04061}$ and *cp190*$^{P11/H31-2}$ mutants (Fig 2E and 2F), suggesting a relationship between γH2Av and *gypsy* insulator components in terms of recruitment to chromatin. Notably, the reduction of γH2Av in *su(Hw)*$^{e04061}$ was more extreme than in other mutants. From this we postulate that the interaction between γH2Av and Su(Hw) insulator sites may be largely mediated through interactions with Su(Hw) itself, although interactions with Mod(mdg4)67.2 and CP190 may also contribute to the overall stability of the complex. These results suggest that Su(Hw) and CP190, but not Mod(mdg4)67.2, are necessary for sustaining normal γH2Av levels.

Our results to this point showed a strong correlation between *gypsy* insulators and γH2Av in chromosomes, but it remained unclear if there was a functional relationship. One key finding is the partial rescue of the *ct*$^6$ phenotype in *su(Hw)*$^{e04061}$/*His2Av*$^{810}$ double heterozygotes (Fig 7). Significant increases in pigmentation in the abdomens of male flies were also found in the *y*$^2$ background, and the *ct*$^6$ phenotype rescue was replicated in the *su(Hw)*$^V$/*His2Av*$^{810}$ background (Fig 7). This demonstrates that H2Av influences the activity of insulator complexes in multiple tissues. Our findings point to a model in which *gypsy* insulator components and γH2Av stabilize each other in the chromatin (Fig 8). Further biochemical analysis will be required to determine if γH2Av is in direct physical contact with Su(Hw) and, if so, which domains are the contact points between these proteins. We also show that γH2Av localizes to insulator bodies and that inhibiting dephosphorylation prevents dissolution of the bodies in experiments inducing recovery after osmotic stress. The mechanism of insulator body formation remains unknown, but our results suggest a model in which γH2Av increases the stability of insulator proteins both at the level of chromatin and at the level of insulator bodies. In this model, dephosphorylation may be required to allow for disassembly of insulator protein complexes (Fig 8A and 8B).

One important consideration about the role of γH2Av in insulator function is whether DNA repair mechanisms are involved in insulator function mechanisms or vice versa. Indeed, it was recently described that CTCF sets the boundaries for phosphorylated H2AX spreading in human cell culture [83] and that CTCF and γH2AX are both recruited to sites of DSBs in mouse embryonic fibroblasts (MEFs) [31]. The link between DNA damage repair and insulator proteins is further corroborated by the recent finding that TAD boundary strength and CTCF insulation strength both increase in an ATM-dependent manner in human cell culture

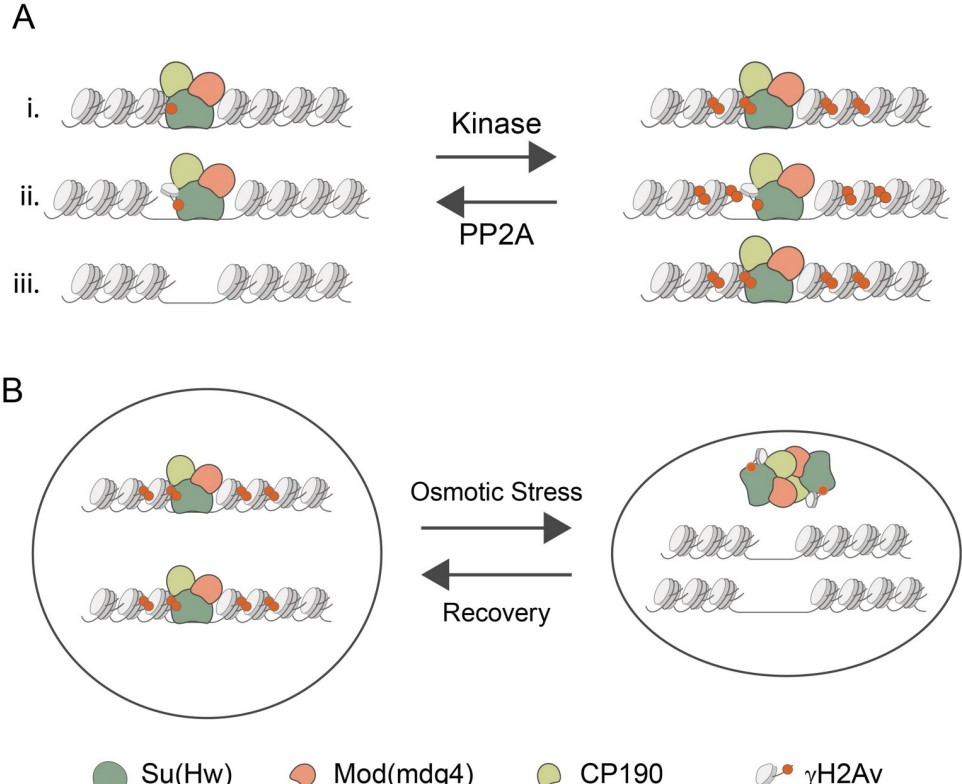

**Fig 8. A proposed model illustrating the role of interactions between *gypsy* insulator components and γH2Av. A.** We speculate that binding of the Su(Hw) insulator complex promotes phosphorylation of neighboring H2Av. This stabilizes binding of the insulator complex and promotes spread of phosphorylated H2Av through neighboring nucleosomes. (i.) Case in which Su(Hw) complexes interact with nucleosomal γH2Av. (ii.) Case in which Su(Hw) complexes interact with non-nucleosomal γH2Av. (iii.) Case in which Su(Hw) is absent and γH2Av is significantly reduced. **B.** During osmotic stress, insulator complexes leave chromatin and form insulator bodies. These bodies contain γH2Av that must be dephosphorylated during recovery so that insulator proteins can return to their genomic binding sites. The mechanism by which γH2Av interacts with Su(Hw) insulator components is still unknown. In addition to direct interactions γH2Av may also interact with insulator proteins through weak multivalent interactions [89].

after X-ray-induced DNA damage [84]. In *Drosophila*, mutation of CTCF is associated with instability in ribosomal DNA found in the nucleolus [85]. Importantly, Su(Hw) has been shown to affect genome stability [25] and we recently described signatures of genome instability during oogenesis in *su(Hw)* mutants as well as the presence of chromosomal aberrations in actively dividing neuroblasts of *su(Hw)*-deficient larvae [24]. In ovaries, the mutation of Su(Hw) produces an increase in the levels of γH2Av, which is in apparent contradiction with the observation in this work that mutation in Su(Hw) leads to a significant reduction in γH2Av in salivary glands. We interpret these results not as a contradiction but as a reflection of how two different tissues respond to mutation in Su(Hw). In ovaries (and developing brain) lack of Su(Hw) leads to genome instability causing DNA breaks that activate DNA repair mechanisms that require γH2Av. In salivary glands we do not have evidence that mutation of Su(Hw) induces genetic instability and this possibility is supported by our observation and the levels of γH2Av are actually reduced in the mutant. These observations suggest the interesting possibility that there are at least two different pathways of γH2Av introduction in *Drosophila*, Su(Hw)-dependent and independent. Our results suggest that the Su(Hw)-dependent γH2Av plays a role in insulator and boundary function, whereas the Su(Hw)-independent would play

its traditional role in DNA repair. How Su(Hw) mutations induce genomic instability remains to be elucidated. However, if the relationship between insulator proteins and the DNA damage repair pathway is conserved from flies to humans, testing for interdependence of these components may provide clinically useful information.

One caveat when addressing these considerations is the dual nature of the histone variant H2Av in *Drosophila*, where it functions as a homolog of the mammalian H2AX, but also as the homolog of the highly conserved histone variant H2AZ [34]. While mammalian H2AX plays a role in DNA repair and genome stability, H2AZ has an important contribution to chromatin structure at promoters of eukaryote genes in general. In *Drosophila*, phosphorylation of H2Av has also been directly implicated in a number of instances where H2Av participates in regulation of chromatin structure [86]. A direct link between γH2Av and transcription, however, was reported showing that γH2Av facilitates PARP1 activity at transcriptionally activated promoters [87]. Conversely, recent results from mammalian cell culture studies highlight a functional interaction between H2AZ, and CTCF, the sole insulator protein in mammals [88]. Taken together, our results reinforce the notion of a strong functional connection between histone H2A variants and insulator function and provide a novel foundation for understanding how the interplay between chromatin insulators and histones influence gene regulation and genome stability. Future experiments are necessary to discern whether the interaction of γH2Av with insulator proteins is a reflection of the existence of DNA repair elements associated with insulator function, as one would expect from a H2AX role, or whether γH2Av works as H2AZ instead, and adds regulatory elements of transcription to chromatin insulator function.

The mechanisms by which γH2Av interacts with insulator proteins is another question that remains to be explored. A direct interaction between the phosphorylated H2Av, but not the unphosphorylated form, with one or more *gypsy* insulator proteins could explain the findings presented here, both at the chromatin level and insulator bodies. Alternatively, research from our lab [89] has revealed that insulator proteins have liquid-liquid phase separation properties *in vivo*, which suggest the possibility that the association between γH2Av and insulator proteins may be mediated by weak multivalent interactions leading to the formation of insulator protein condensates.

## Supporting information

**S1 Fig. Additional examples of Peak profiles of insulator protein Su(Hw) compared with γH2Av, H2Av Homotypic nucleosome distribution and TAD boundaries.** Profiles include examples from all chromosomes and clusters, as defined in Fig 3.
(TIF)

**S2 Fig. Treatment with okadaic acid increases the amount of phosphorylated H2Av on polytene chromosomes.** Shown are co-immunostains of polytene chromosomes from salivary glands treated with okadaic acid. **A.** Immunostaining of γH2Av with Su(Hw). **B.** Quantification of the immunostaining data show in A. The intensities of Su(Hw) and γH2Av are shown bellow, in the absence and presence of okadaic acid. Each point represents the polytene genome of an individual cell. Error bars represent one standard error of the mean. P-values were determined using unpaired two-tailed Student's T-tests. **C.** Immunostaining of γH2Av with Mod(mdg4)67.2. **D.** Immunostaining of γH2Av with CP190. Immunofluorescent micrographs of polytene chromosome squashes are shown on the left. Magnified insets are shown to the right of each figure, corresponding to the white boxes in the figures on the left. Scale bars are 5 μm in the figures and 2 μm in the insets. Insets are shown as RGB merge, with DAPI on the blue channel, γH2Av on the green channel, and various insulator proteins on the red

channel. Red and green channels are shown independently in grey scale and merged as magenta and green. Beneath the insets are linescans corresponding to the yellow lines in the merged insets. Linescan intensities in A, C, and D were normalized by dividing each value by the maximum intensity recorded on each channel.
(TIF)

**S3 Fig. Phosphatase inhibition restores localization of γH2Av at insulator sites in insulator protein mutants.** Shown are co-immunostains of polytene chromosomes from salivary glands treated with okadaic acid. **A.** Colocalization of γH2Av with Mod(mdg4)67.2 in *su(Hw)$^{e04061}$*. **B.** Colocalization of γH2Av with CP190 in *su(Hw)$^{e04061}$*. **C.** Colocalization of γH2Av with Su(Hw) in *mod(mdg4)$^{u1}$*. **D.** Colocalization of γH2Av with CP190 in *mod(mdg4)$^{u1}$*. **E.** Colocalization of γH2Av with Su(Hw) in *cp190$^{P11/H31-2}$*. **F.** Pearson's Correlation Coefficient (PCC) for γH2Av signal with each insulator protein signal is plotted, with each point representing the polytene genome of each cell. Error bars represent one standard error of the mean. PCC values are grouped by genotype (red = *su(Hw)$^{e04061}$*, green = *mod(mdg4)$^{u1}$*, blue = *cp190$^{P11/H31-2}$*). Immunostaining results from polytene chromosome squashes are shown on the left in each panel. Magnified insets are shown to the right of each figure, corresponding to the white boxes in the figures on the left. Scale bars are 5 μm in the figures and 2 μm in the insets. Insets are shown as RGB merge, with DAPI on the blue channel, γH2Av on the green channel, and various insulator proteins on the red channel. Red and green channels are shown independently in grey scale and merged as magenta and green. Beneath the insets are linescans corresponding to the yellow lines in the merged insets. Linescan intensities were normalized by dividing each value by the maximum intensity recorded on each channel.
(TIF)

## Acknowledgments

We would like to thank former members of the Labrador lab, including Dr. Emily Stow, for collaboration and discussion. We thank Dr. Rachel Patton McCord for critical review of the manuscript. Stocks obtained from the Bloomington *Drosophila* Stock Center (NIH P40OD018537) were used in this study. S2 cell culture was obtained from the *Drosophila* Genomics Resource Center (NIH 2P40OD010949). This publication is dedicated to the memory of Ran An, a co-first author of the paper that recently passed away victim of cancer at a very early age. Her memory will remain in our hearts forever.

## Author Contributions

**Conceptualization:** James R. Simmons, Ran An, Mariano Labrador.

**Data curation:** James R. Simmons, Ran An.

**Formal analysis:** James R. Simmons, Bright Amankwaa, Mariano Labrador.

**Funding acquisition:** Mariano Labrador.

**Investigation:** James R. Simmons, Bright Amankwaa, Shannon Zayac, Justin Kemp, Mariano Labrador.

**Methodology:** James R. Simmons, Ran An, Mariano Labrador.

**Project administration:** Mariano Labrador.

**Resources:** Mariano Labrador.

**Software:** James R. Simmons.

**Supervision:** Mariano Labrador.

**Validation:** James R. Simmons.

**Visualization:** James R. Simmons, Bright Amankwaa, Mariano Labrador.

**Writing – original draft:** James R. Simmons, Bright Amankwaa, Mariano Labrador.

**Writing – review & editing:** James R. Simmons, Mariano Labrador.

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
