## [Decision Letter · Decision Letter 0]

27 Apr 2022

Dear Dr Labrador,

Thank you very much for submitting your Research Article entitled 'Phosphorylated Histone variant γH2Av is associated with Chromatin Insulators in Drosophila' to PLOS Genetics.

The manuscript was seen by reviewers who previously evaluated it at review commons. As you will see, the reviewers appreciated the attention to an important problem, but raised some substantial concerns about the current manuscript. Based on the reviews, we will not be able to accept this version of the manuscript, but we would be willing to review a much-revised version. We cannot, of course, promise publication at that time.

If you decide to revise the manuscript for further consideration at PLOS Genetics, please aim to resubmit within the next 60 days, unless it will take extra time to address the concerns of the reviewers, in which case we would appreciate an expected resubmission date by email to plosgenetics@plos.org.

[LINK]

We are sorry that we cannot be more positive about your manuscript at this stage. Please do not hesitate to contact us if you have any concerns or questions.

Yours sincerely,

Gregory S. Barsh

Editor-in-Chief

PLOS Genetics

Gregory Copenhaver

Editor-in-Chief

PLOS Genetics

Reviewer's Responses to Questions

**Comments to the Authors:**

Reviewer #1: Presented study is aimed to describe a complex relationships between Drosophila architectural protein Su(Hw) and γH2Av histone marker. Authors describe an existence of two functionally different γH2Av pools in Drosophila cells. They link the first one with the DNA repair mechanisms: strong foci which can be detected in Drosophila tissues or cells with any γH2Av antibodies both via immunostaining or via ChIP-Seqs. The second γH2Av pool (linked by Authors to transcriptional regulation) exists in untreated diploid cells at normal conditions and can be detected only with highly-sensitive anti-γH2Av antibodies mainly by ChIP-Seqs methods.

A complex nature of relationships between Su(Hw) and γH2Av was revealed by Authors. In their previous articles they describe a link between Su(Hw) and γH2Av in ovaries and neuroblasts (Hsu et al., 2020). A strong increase was detected in Su(Hw) mutants via immunostaining with anti- γH2Av antibodies. Authors linked this increase in γH2Av modification level to the DNA damage repair mechanisms.

In the present study, the authors describe the reverse effect of the Su(Hw) mutation at the level of the γH2Av modification. They demonstrated a decrease in the γH2Av level at the polythene chromosomes of the salivary glands. Authors link γH2Av pool at the untreated salivary glands nucleus to some transcriptional functions of the γH2Av.

In my opinion, in order to support their claims, the authors need to provide evidence of the loss of the transcriptionally linked γH2Av pool when the conditions are violated (Su(Hw) mutated or phosphatase inhibitor added) via some sensitive method (for example by ChIP-Seq). Authors wrote that they decided against to use in the current research previously published salivary glands γH2Av ChIP-Seq GSE55932, PMID: 25437540 due to their suspicion to antibodies quality. However, they stated that they have more sensitive, better quality anti-γH2Av antibodies. Thus, having good antibodies, they are able to perform anti-γH2Av ChIP-Seq analysis under disturbed conditions. It can be performed in salivary glands in a mutated background or in S2 cells with Su(Hw) knockdown.

One more additional experiment will improve the quality of the manuscript – Co-IP experiment between Su(Hw) and γH2Av. In fact, such an experiment was requested by two of three Reviewers during the previous round of the Review. Authors wrote that there is no need in such an experiment as they are not imply on an existence of a direct interaction between Su(Hw) and γH2Av. But the benefit of the Co-IP that it reveals not only direct but also indirect (mediated with some other protein) interaction between investigated proteins. So, it would be of real help here. Moreover, in the abstract of the manuscript Authors continue to state that “we demonstrate an interaction between proteins that associate with the gypsy insulator and the phosphorylated histone variant H2Av (γH2Av)”. In order to wrote such a statement, they should support it with experimental evidence of Co-IP.

Minor points

1) In the Discussion Authors referred their previous work describing impact of Su(Hw) on DNA damage in ovaries and neuroblasts. They avoid direct references to the fact that in ovaries Su(Hw) mutation led to the increase in the level of the γH2Av. I think this fact is very important and should be properly discussed, even though it goes against the ideas of the current manuscript. It demonstrates that there are at least two different pathways of γH2Av introduction in Drosophila (the Su(Hw)-dependent and independent).

2) In the reply to the Reviewers Authors stated that they added a western blot panel at the Fig 6. But I can not find such a panel in the final version of the article

Conclusion:

Presented manuscript provides a previously undescribed link between γH2Av modification, known for its role in DNA damage repair, and Su(Hw) architectural protein. In my opinion, some additional experiments should be provided by Authors to support their statements. Particularly, ChIP-Seq experiments by anti- γH2Av antibodies at the impaired conditions (by SuHw mutation or knockdown). This experiment brings causation to the provided research. The Co-IP experiment between Su(Hw) and γH2Av will also benefit the current study by removing suspicions about γH2Av antibodies.

In the previous round of the Review Authors stated that these requested experiments are out of scope of the current research. But I feel in opposite way. I believe that PLOS Genetics Journal is a well-known journal with a wide audience, worthy of publications based on thorough experimental data.

Reviewer #2: One of my major concerns of the original manuscript is the "use of γH2Av immunostaining interchangeably with the action of phosphorylation of H2Av". The claim is not supported without experimental evidence measuring the ratio between γH2Av/H2Av/H2A, at least in key experiments. In the revised manuscript, the authors plan to focused on their "identifying a role for γH2Av in insulator function" and "avoid referring to phosphorylation of H2Av as part of a mechanism and have replaced it, when possible, by phosphorylated H2Av through the text." However, a quick scan still found some of these inappropriate claims. For example, on page 13, "reduced levels of H2Av phosphorylation" was used twice in reference to reduced level of γH2Av. Similar statements are also found in multiple places from page 15-17 and in the discussion. These should be revised when appropriate.

The authors have addressed my other concerns.

Reviewer #3: The authors have appropriate addressed my questions and suggestions in the proposed revision and I do not have further concerns about the work.

**Have all data underlying the figures and results presented in the manuscript been provided?**

Reviewer #1: Yes

Reviewer #2: Yes

Reviewer #3: None

PLOS authors have the option to publish the peer review history of their article (what does this mean?). If published, this will include your full peer review and any attached files.

Reviewer #1: No

Reviewer #2: No

Reviewer #3: No

---

## [Decision Letter · Decision Letter 1]

24 Aug 2022

Dear Dr Labrador,

We are pleased to inform you that your manuscript entitled "Phosphorylated Histone variant γH2Av is associated with Chromatin Insulators in Drosophila" has been editorially accepted for publication in PLOS Genetics. Congratulations!

The revised manuscript was seen by two of the previous reviewers; as you will see, both are positive.

Yours sincerely,

Gregory S. Barsh

Editor-in-Chief

PLOS Genetics

Gregory Copenhaver

Editor-in-Chief

PLOS Genetics

Comments from the reviewers (if applicable):

Reviewer's Responses to Questions

**Comments to the Authors:**

Reviewer #1: The authors have properly discussed all the controversial points raised in my previous review and included the relevant discussion in the main body of the manuscript. I have no other concerns, so this manuscript can be published.

Reviewer #2: The authors have addressed my critics.

**Have all data underlying the figures and results presented in the manuscript been provided?**

Reviewer #1: Yes

Reviewer #2: Yes

PLOS authors have the option to publish the peer review history of their article (what does this mean?). If published, this will include your full peer review and any attached files.

Reviewer #1: No

Reviewer #2: No

**Data Deposition**

http://datadryad.org/submit?journalID=pgenetics&manu=PGENETICS-D-22-00400R1

**Press Queries**

---

## [Editor Report · Acceptance letter]

22 Sep 2022

PGENETICS-D-22-00400R1 

Phosphorylated Histone variant γH2Av is associated with Chromatin Insulators in Drosophila 

Dear Dr Labrador, 

We are pleased to inform you that your manuscript entitled "Phosphorylated Histone variant γH2Av is associated with Chromatin Insulators in Drosophila" has been formally accepted for publication in PLOS Genetics! Your manuscript is now with our production department and you will be notified of the publication date in due course.

With kind regards,

Zsofi Zombor

PLOS Genetics

On behalf of:
